# An algorithm to quantify intratumor heterogeneity based on alterations of gene expression profiles

Mengyuan Li[1,2,3,4], Zhilan Zhang[1,2,3,4], Lin Li[1,2,3] & Xiaosheng Wang [1,2,3✉]

Intratumor heterogeneity (ITH) is a biomarker of tumor progression, metastasis, and immune evasion. Previous studies evaluated ITH mostly based on DNA alterations. Here, we developed a new algorithm (DEPTH) for quantifying ITH based on mRNA alterations in the tumor. DEPTH scores displayed significant correlations with ITH-associated features (genomic instability, tumor advancement, unfavorable prognosis, immunosuppression, and drug response). Compared to DNA-based ITH scores (EXPANDS, PhyloWGS, MATH, and ABSOLUTE), DEPTH scores had stronger correlations with antitumor immune signatures, cell proliferation, stemness, tumor advancement, survival prognosis, and drug response. Compared to two other mRNA-based ITH scores (tITH and sITH), DEPTH scores showed stronger and more consistent associations with genomic instability, unfavorable tumor phenotypes and clinical features, and drug response. We further validated the reliability and robustness of DEPTH in 50 other datasets. In conclusion, DEPTH may provide new insights into tumor biology and potential clinical implications for cancer prognosis and treatment.

[1] Biomedical Informatics Research Lab, School of Basic Medicine and Clinical Pharmacy, China Pharmaceutical University, Nanjing 211198, China. [2] Cancer Genomics Research Center, School of Basic Medicine and Clinical Pharmacy, China Pharmaceutical University, Nanjing 211198, China. [3] Big Data Research Institute, China Pharmaceutical University, Nanjing 211198, China. [4] These authors contributed equally: Mengyuan Li, Zhilan Zhang. ✉email: xiaosheng.wang@cpu.edu.cn

Genomic instability is a major cause of tumor heterogeneity, which refers to genetic and phenotypic variation within (intratumor heterogeneity (ITH)) and between tumors (intertumor heterogeneity)[1]. Accordingly, many genomic feature-based algorithms have been proposed to quantify tumor heterogeneity, such as ABSOLUTE[2], MATH[3], EXPANDS[4,5], and PhyloWGS[6]. Based on DNA copy number alteration (CNA) profiles, ABSOLUTE evaluates tumor ploidy estimates representing ITH[2]. MATH assesses ITH based on somatic mutation profiles[7]. EXPANDS predicts clonal subpopulations of tumor cells representing ITH based on the proportion of cells with specific mutation profiles[4]. PhyloWGS infers the subclonal composition of tumor cells based on their mutations and CNAs[6]. Besides genome profiles, transcriptome, proteome, and epigenome profiles were also used to define ITH[8]. High ITH is often associated with an unfavorable prognosis in cancer[6]. Moreover, ITH often has a prevalent negative correlation with tumor immunity[9–13]. Currently, cancer immunotherapies, e.g., the immune checkpoint blockade (ICB) and chimeric antigen receptor (CAR) T cell therapies, have demonstrated success in treating diverse cancers[14–16]. Nevertheless, only a subset of cancer patients currently respond to such therapies. Certain predictive biomarkers for the response to ICB have been identified, such as PD-L1 expression[17], tumor mutation burden (TMB)[18], mismatch repair deficiency (dMMR) or microsatellite instability (MSI)[19], and tumor-infiltrating lymphocyte (TIL) levels[20].

Because alterations in genome and epigenome profiles often lead to heterogeneous gene expression profiles in tumors, defining ITH based on gene expression profiles is a viable approach[21–23]. Park et al. proposed a method (tITH) for evaluating transcriptome-based ITH using RNA-Seq data[24]. This method has exhibited certain effectiveness in evaluating ITH, e.g., the positive correlation between its ITH and genetic ITH characterized by DNA-based methods, and the negative correlation between the ITH and survival prognosis in the tumor. Nevertheless, because the tITH method is based on the protein–protein interaction (PPI) network, it is susceptible to PPIs' reliability. Also, tITH cannot evaluate ITH when gene expression profiles in normal samples are not available.

In this study, we proposed the Deviating gene Expression Profiling Tumor Heterogeneity (DEPTH) algorithm for evaluating ITH levels at the mRNA level. By analyzing 25 cancer types from The Cancer Genome Atlas (TCGA) program (https://www.cancer.gov/about-nci/organization/ccg/research/structural-genomics/tcga), we demonstrated that the ITH defined by DEPTH had the common features of ITH characterized in previous studies, such as its negative correlation with tumor prognosis and antitumor immunity[5,9,12] and positive correlation with drug resistance[25]. We further validated these features of ITH evaluated by DEPTH in more than 10,000 tumor samples apart from TCGA. We compared our method with six other ITH evaluation methods, including ABSOLUTE[2], MATH[3], EXPANDS[4,5], PhyloWGS[6], tITH[24], and sITH[26]. Our data showed that DEPTH is an effective and robust method for evaluating ITH, and its performance is superior to or comparable to that of the other methods.

## Results

The DEPTH algorithm shows that in a tumor, when most genes simultaneously display high or simultaneously display low expression deviations from their mean expression values in normal or tumor samples, the tumor will have a low-DEPTH score (low ITH). By contrast, in a tumor, when many genes exhibit high expression deviations, and while many other genes exhibit low expression deviations from their mean expression values, the tumor will have a high DEPTH score (high ITH). Thus, the ITH defined by DEPTH represents the asynchrony of transcriptome alterations in tumor cells in a manner.

**In silico proof of concept of DEPTH scores**. We performed the in silico simulation to examine whether DEPTH scores indeed represent ITH in tumors. We created gene expression profiles of 202, 40, and 1375 simulated tumor samples based on three cell line or single-cell RNA-Seq (scRNA-seq) datasets (Cancer cell lines[27], GSE69405[28], GSE113660[29,30]), respectively. We defined the expression value of a gene in a simulated sample as the maximum expression value of the gene in all cells constituting the sample. Based on the gene expression profiles, we calculated the DEPTH score of each simulated tumor samples. We observed strong correlations between the DEPTH scores of the simulated tumor samples and the numbers of cell lines or single cells they contained ($\rho = 0.94$, 0.97, 0.998, respectively) (Fig. 1a). These results indicate that the more heterogeneous tumor samples (composed of more different cells) have higher DEPTH scores. Moreover, based on the cancer cell line dataset[27], we produced 10 sets of simulated tumor samples with each set of simulated samples composed of an equal number ($m$) of cell lines. We found that the DEPTH scores had a strong positive correlation with the numbers of different cancer types the cell lines originate from in the simulated tumor samples ($\rho = 0.999$, 0.999, 1, 1, 0.999, 0.999, 1, 1, 0.999, and 1 for $m = 62$, 49, 46, 42, 35, 32, 31, 30, 29, and 27, respectively) (Fig. 1b). We performed a similar experiment in the scRNA-seq dataset GSE57872[31]. Likewise, we observed a strong positive correlation between the DEPTH scores and the numbers of different cell lines the single cells belonged to in the simulated tumor samples ($\rho = 1$, 1, 0.952, 0.952, 0.881, 0.976, and 0.810 for $m = 44$, 58, 65, 70, 73, 75, and 94, respectively). These results indicate that the more heterogeneous tumor samples (composed of more different types of cell lines or cancers) have higher DEPTH scores. Collectively, these experiments demonstrate that DEPTH scores indeed represent ITH in tumors.

**Association of DEPTH scores with genomic instability**. Genomic instability often results in increased TMB[32]. We found that TMB was positively correlated with DEPTH scores in pan-cancer ($p = 1.26 \times 10^{-115}$, $\rho = 0.26$) and in 11 individual cancer types (false discovery rate (FDR)-adjusted Spearman's correlation test $p$ ($FDR_{sp}$) < 0.05) (Fig. 2a). The cancer types with higher TMB tended to exhibit higher DEPTH scores than those with lower TMB. For example, SKCM, which had significant higher TMB than PRAD ($p = 7.61 \times 10^{-116}$; median TMB: 461.5 versus 52), displayed higher DEPTH scores than PRAD ($p = 5.18 \times 10^{-160}$; median DEPTH score: 17.73 versus 2.95). p53 plays a crucial role in maintaining genomic stability[33]. We found that TP53-mutated tumors exhibited remarkably higher DEPTH scores than TP53-wildtype tumors in pan-cancer ($p = 2.21 \times 10^{-13}$) and in multiple individual cancer types, including BLCA, BRCA, LIHC, LUAD, LUSC, PRAD, STAD, and UCEC (FDR < 0.05) (Fig. 2b). Also, DNA mismatch repair deficiency (dMMR) or microsatellite instability (MSI) is a prevalent pattern of genomic instability[32]. We found that MSI-high (MSI-H) tumors displayed significantly higher DEPTH scores than MSI-low (MSI-L) or microsatellite stability (MSS) tumors in the cancers with a high prevalence of MSI, including COAD, STAD, and ESCA ($p < 0.05$) (Fig. 2c). Moreover, we observed that three DNA mismatch repair proteins (PCNA, MSH6, and MSH2) showed significant positive expression correlations with DEPTH scores in at least 8 cancer types (two-sided Student's $t$ test, FDR < 0.05) (Fig. 2d). We found many DNA damage response-associated pathways that were more

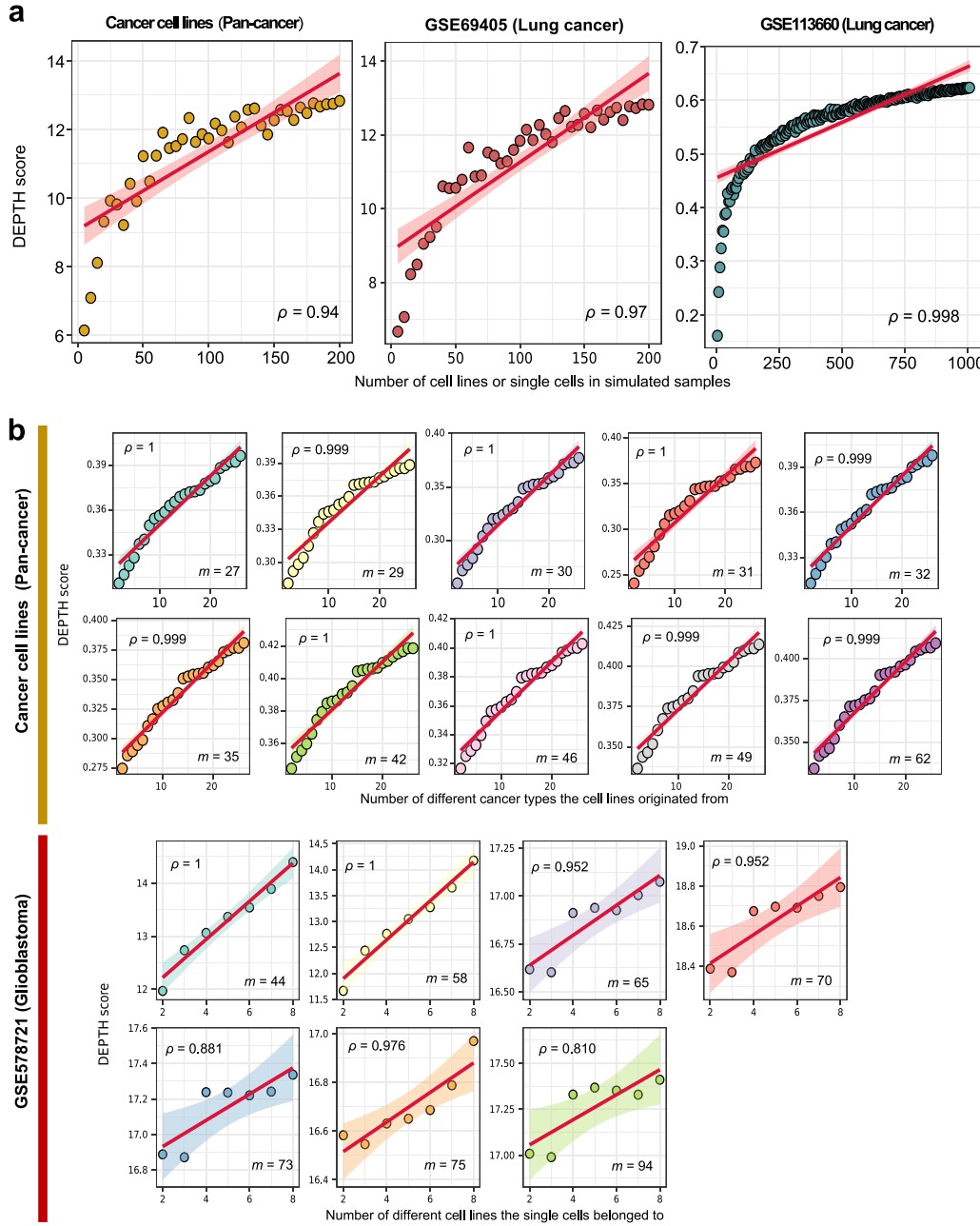

**Fig. 1 In silico simulation experiments using four cell line or single-cell RNA-Seq datasets. a** Correlations between DEPTH scores and the numbers of cell lines or single cells in the simulated tumor samples. **b** Correlations between DEPTH scores and the numbers of different cancer types the cell lines originated from or cell lines the single cells belonged to in the simulated tumor samples. $\rho$, Spearman correlation coefficient. It also applies to the following figures.

highly enriched in high-DEPTH-score than in low-DEPTH-score tumors in at least five cancer types by GSEA[34]. These pathways included DNA replication, base excision repair, homologous recombination, and mismatch repair (Fig. 2e).

In a recent study[35], Knijnenburg et al. identified functionally deleterious mutations in the genes in nine major DNA damage repair (DDR) pathways in TCGA pan-cancer. The nine DDR pathways included mismatch repair, base excision repair, nucleotide excision repair, the Fanconi anemia (FA) pathway, homology-dependent recombination, non-homologous DNA end joining, direct damage reversal/repair, translesion DNA synthesis, and damage sensor. Based on the mutations in the genes in the DDR pathways, we classified pan-cancer into pathway-gene-wildtype and pathway-gene-mutated groups for each of the nine

DDR pathways. The pathway-gene-wildtype indicates no functionally deleterious mutations in the pathway genes, and the pathway-gene-mutated indicates at least a functionally deleterious mutation in the pathway genes. Strikingly, we found that DEPTH scores were significantly higher in the pathway-gene-mutated group than in the pathway-gene-wildtype group consistently for the nine DDR pathways ($p < 0.05$) (Fig. 2f). These results indicate that the enhanced DEPTH scores are associated with DDR deficiency, again demonstrating the positive association of DEPTH scores with genomic instability. Homologous recombination deficiency (HRD) may lead to large-scale genomic instability[35]. Knijnenburg et al. evaluated HRD scores for 9125 TCGA cancer samples by combining the scores of HRD loss of heterozygosity, large-scale state transitions, and the number of

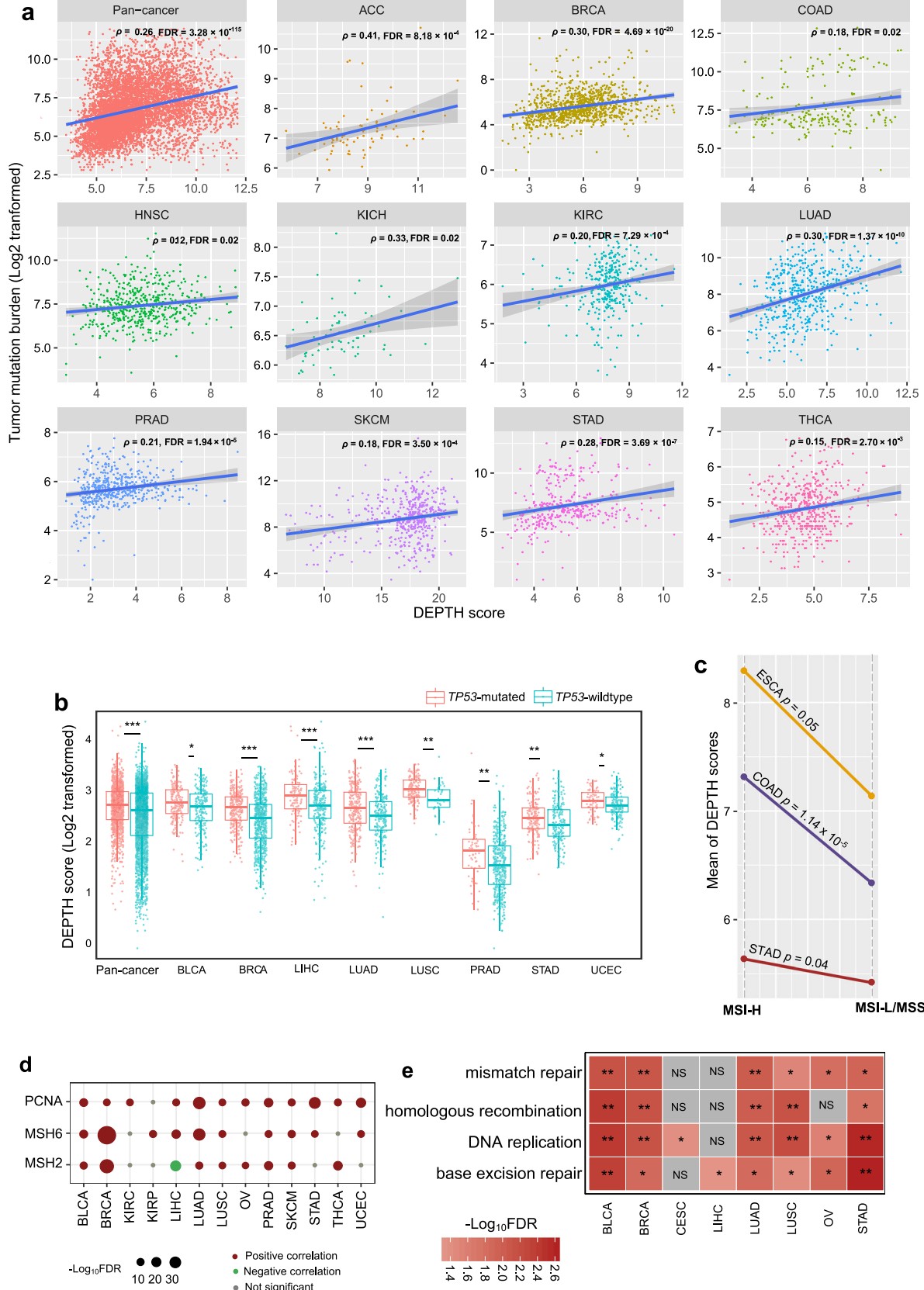

**Fig. 2** (Continued)

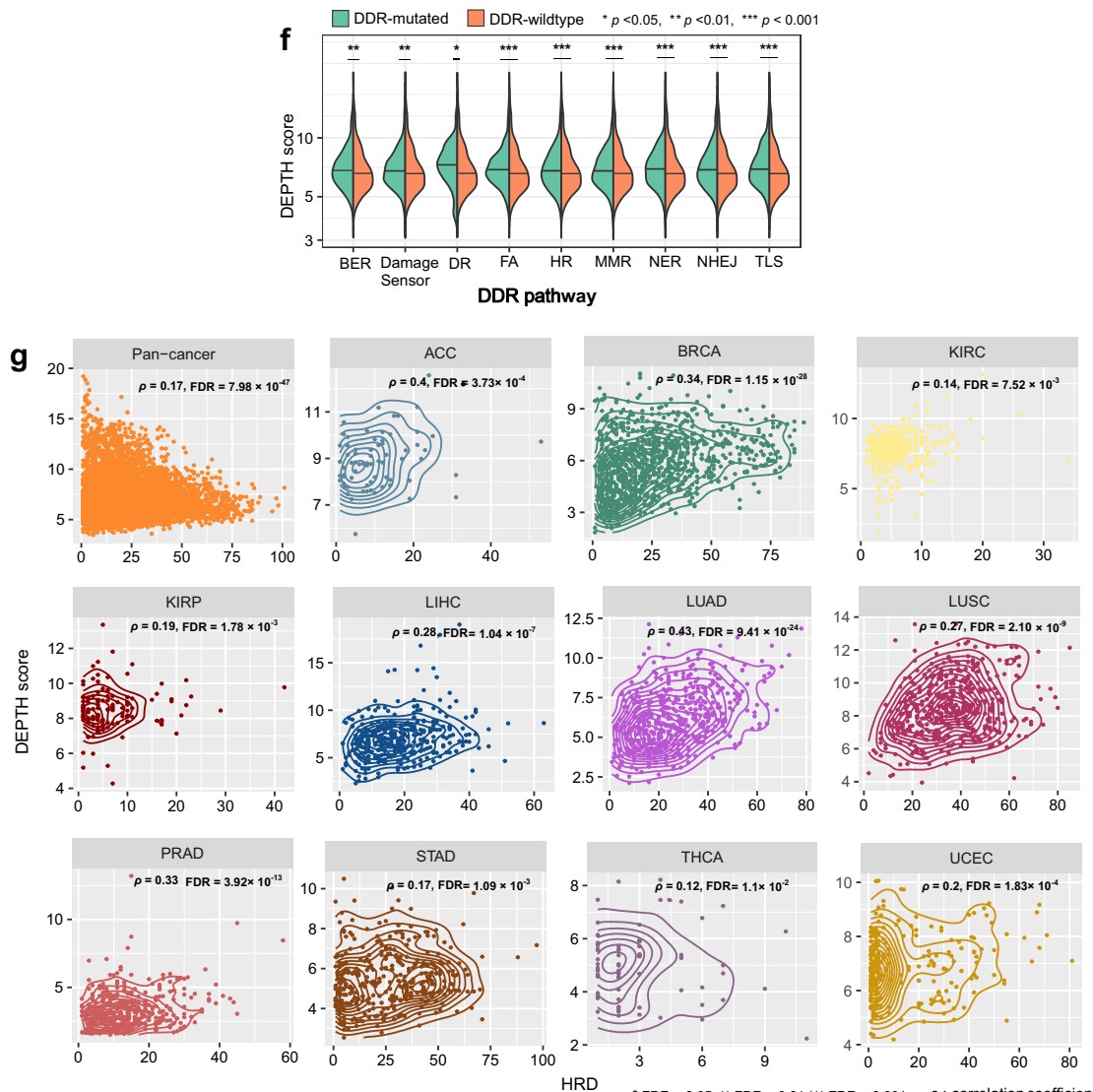

**Fig. 2 Association of DEPTH scores with genomic instability. a** The positive correlations between tumor mutation burden (TMB) and DEPTH scores in pan-cancer and in 11 individual cancer types. Spearman's correlation test, false discovery rate (FDR), and correlation coefficient ($\rho$) are shown. The FDR was estimated by the Benjamini and Hochberg method[66] to adjust for $p$-values in multiple tests. TMB, the total somatic mutation count in the tumor. **b** *TP53*-mutated tumors display significantly higher DEPTH scores than *TP53*-wildtype tumors in pan-cancer and in multiple individual cancer types (FDR < 0.05). **c** MSI-high (MSI-H) tumors have significantly higher DEPTH scores than MSI-low (MSI-L)/microsatellite stability (MSS) tumors in COAD, STAD, and ESCA ($p \leq 0.05$). MSI, microsatellite instability. **d** The positive associations between the expression of DNA mismatch repair proteins (PCNA, MSH6, and MSH2) and DEPTH scores within multiple individual cancer types (two-sided Student's $t$ test, FDR < 0.05). **e** The four DNA damage response-associated pathways more highly enriched in high-DEPTH-score than in low-DEPTH-score tumors within multiple individual cancer types identified by GSEA[34] (FDR < 0.05). high-DEPTH-score, DEPTH scores in the upper third. low-DEPTH-score, DEPTH scores in the bottom third. They also apply to the following figures. **f** Comparison of DEPTH scores between the pathway-gene-mutated and pathway-gene-wildtype groups for nine DDR pathways. DDR, DNA damage repair. BER, base excision repair. MMR, mismatch repair; BER, base excision repair. NER, nucleotide excision repair. FA, Fanconi Anemia. HDR, homology-dependent recombination. NHEJ, non-homologous DNA end joining. DR, direct damage reversal/repair. TLS, translesion DNA synthesis. **g** The positive correlation between DEPTH scores and HRD scores in pan-cancer and in 11 individual cancer types. HRD, homologous recombination deficiency. *FDR < 0.05, **FDR < 0.01, ***FDR < 0.001, NS: not significant. It also applies to the following figures.

telomeric allelic imbalances. We found that DEPTH scores had a significant positive correlation with HRD scores in pan-cancer and 11 individual cancer types (FDR$_{Sp}$ < 0.05) (Fig. 2g).

Altogether, these data suggest that the DEPTH level has a strong positive association with genomic instability in cancer.

**Associations of DEPTH scores with clinical characteristics.** ITH is associated with poor prognosis in cancer[36]. Survival analyses showed that higher DEPTH scores were associated with worse survival in pan-cancer (log-rank test, $p = 3.31 \times 10^{-38}$, $2.88 \times 10^{-31}$, $7.4 \times 10^{-14}$, $2.32 \times 10^{-50}$ for overall survival (OS), disease-specific survival (DSS), disease-free interval (DFI), and progression-free interval (PFI), respectively.) (Fig. 3a). Also, in 5 individual cancer types (BRCA, COAD, KIRC, LUAD, and UCEC), higher DEPTH scores were associated with worse OS (log-rank test, $p < 0.05$) (Supplementary Fig. 1a). Among them, BRCA, COAD, LUAD, and KIRC are highly prevalent cancer types.

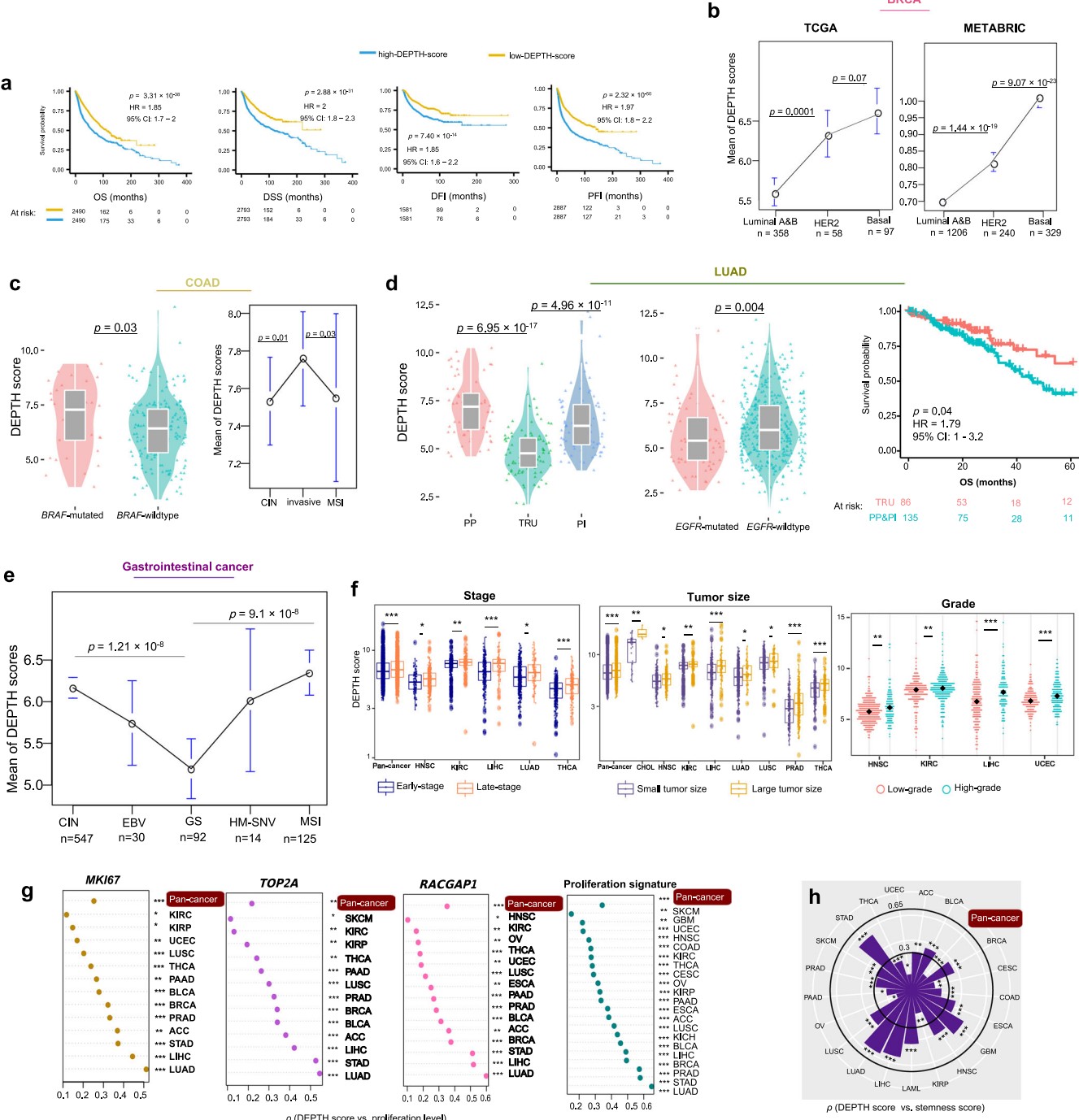

**Fig. 3 Association of DEPTH scores with clinical characteristics. a** Kaplan−Meier survival curves displaying the negative correlation between DEPTH scores and survival (OS, DSS, PFI, and DFI) in pan-cancer. The log-rank test *p*-values, hazard ratio (HR), and 95% confidence interval (CI) are shown. OS, overall survival. DSS, disease-specific survival. PFI, progression-free interval. DFI, disease-free interval. **b** Comparison of DEPTH scores between different breast cancer subtypes (luminal A&B, HER2-enriched, and basal-like) in TCGA and METABRIC datasets. **c** DEPTH scores are significantly higher in *BRAF*-mutated than in *BRAF*-wildtype COAD and are significantly higher in the invasive than in the MSI and CIN subtypes of COAD. CIN, chromosomal instability. **d** DEPTH scores are significantly lower in the TRU than in the PP and PI subtypes of LUAD, and TRU has a favorable overall survival than the PP and PI subtypes; DEPTH scores are significantly lower in *EGFR*-mutated than in *EGFR*-wildtype LUAD. TRU, terminal respiratory unit. PI, proximal-inflammatory. PP, proximal-proliferative. **e** DEPTH scores are significantly higher in MSI and CIN than in GS GI cancers. GS, genome stable. GI, gastrointestinal. HM-SNV, hypermutated-SNV. EBV, Epstein-Barr virus. **f** DEPTH scores increase with tumor progression in pan-cancer and in multiple individual cancer types. The DEPTH scores between late-stage (stage III-IV) and early-stage (stage I-II), between large tumor size (T3-4) and small tumor size (T1-2), and between high-grade (G3-4) and low-grade (G1-2) tumors were compared. The one-sided Mann–Whitney U test FDR < 0.05 indicates the statistical significance. **g** The positive correlations of DEPTH scores with the expression levels of cell proliferation marker genes (*MKI67*, *TOP2A*, and *RACGAP1*) and proliferation signature scores in pan-cancer and in multiple individual cancer types (FDR$_{sp}$ < 0.05). **h** The positive correlations between DEPTH scores and tumor stemness scores in pan-cancer and in 20 individual cancer types (FDR$_{sp}$ < 0.05). Proliferation signature and tumor stemness scores were calculated by the single-sample gene-set enrichment analysis[34]. FDR$_{sp}$, Spearman's correlation test adjusted *p*-value (FDR). It also applies to the following figures.

We further correlated DEPTH scores with clinical features within and across subtypes of several prevalent cancer types, including BRCA, COAD, LUAD, and gastrointestinal (GI) pan-cancer. In BRCA, we compared DEPTH scores between triple-negative breast cancer (TNBC) and non-TNBC, which were classified based on immunohistochemical (IHC) testing. We found that DEPTH scores were significantly higher in TNBC than in non-TNBC ($p = 8.24 \times 10^{-9}$). Furthermore, we found that DEPTH scores were negatively associated with OS in non-TNBC (log-rank test, $p = 0.002$) but not in TNBC ($p = 1$). A possible explanation is that the variation of DEPTH scores was much lower in TNBC than in non-TNBC (2.07 versus 3.09) so that the high-DEPTH-score and low-DEPTH-score tumor samples were not separated in TNBC as clearly as in non-TNBC. Also, we compared DEPTH scores between luminal A&B (ER+), HER2-enriched, and basal-like breast cancer subtypes, which were determined by the PAM50 assay[37]. We found that DEPTH scores were significantly lower in luminal A&B than in HER2-enriched or basal-like subtype ($p < 0.001$) and that DEPTH scores were lower in HER2-enriched versus basal-like subtype ($p = 0.07$) (Fig. 3b). We observed similar results in another large-scale breast cancer genomics dataset METABRIC[38] (Fig. 3b). Furthermore, DEPTH scores displayed a significant negative correlation with OS in luminal A&B (log-rank test, $p = 0.005$) but not in HER2-enriched or basal-like subtype ($p > 0.1$). Again, the possible explanation is that the variation of DEPTH scores was much higher in luminal A&B than in HER2-enriched or basal-like subtype (2.86 versus 1.26 or 2.06). Overall, these results indicate that DEPTH scores conform to the prognostic risk in breast cancer in that the luminal A&B subtype often has a better prognosis than the HER2-enriched and basal-like/TNBC subtypes and the basal-like/TNBC subtype has the worst prognosis among all breast cancer subtypes[39]. Interestingly, we found that DEPTH scores were markedly lower in luminal A than in luminal B subtype ($p = 8.31 \times 10^{-20}$). This result again suggests the positive association between DEPTH scores and breast cancer risk since luminal A cancers have the best prognosis, and luminal B cancers tend to develop slightly faster and have a somewhat worse prognosis than luminal A cancers[40].

In COAD, we compared DEPTH scores between *BRAF*-mutated and *BRAF*-wildtype tumors and found that the *BRAF*-mutated COAD had significantly higher DEPTH scores than *BRAF*-wildtype COAD ($p = 0.03$) (Fig. 3c). This result suggests that the elevated DEPTH scores are associated with worse clinical outcomes in light of the significant association between *BRAF* mutations and poorer prognosis in COAD[41]. When comparing DEPTH scores between three mRNA subtypes (MSI, chromosomal instability (CIN), invasive) of COAD[42], we found that the invasive subtype had higher DEPTH scores than the other two subtypes ($p < 0.05$) (Fig. 3c), again suggesting a significant association between high DEPTH scores and unfavorable clinical outcomes in COAD. Furthermore, survival analyses showed that the elevated DEPTH scores were associated with worse OS trends in non-MSI and MSI COAD (Supplementary Fig. 1b). In LUAD, we compared DEPTH scores between three transcriptional subtypes: terminal respiratory unit (TRU), proximal-inflammatory (PI), and proximal-proliferative (PP). We found that DEPTH scores were the lowest in TRU tumors ($p = 6.95 \times 10^{-17}$, $4.96 \times 10^{-11}$ for TRU versus PP and TRU versus PI, respectively) (Fig. 3d). These results suggest a negative association between DEPTH scores and clinical outcomes in LUAD since TRU is prognostically favorable compared with the other subtypes (Fig. 3d). The *EGFR* mutation is associated with a more favorable prognosis in lung cancer[43,44]. We found that *EGFR*-mutated LUAD had lower DEPTH scores than *EGFR*-wildtype LUAD ($p = 0.004$) (Fig. 3d). Again, this result suggests a negative

association between DEPTH scores and prognosis in LUAD. The GI pan-cancer included GI tract adenocarcinomas (GIACs) composed of 79 esophageal, 383 gastric, 341 colon, and 118 rectal cancers, which were classified into five subtypes: Epstein-Barr virus (EBV), MSI, hypermutated-SNV (HM-SNV), CIN, and genome stable (GS)[45]. DEPTH scores were significantly higher in MSI and CIN versus GS tumors ($p = 1.21 \times 10^{-8}$, $9.10 \times 10^{-8}$ for MSI versus GS and CIN versus GS, respectively) (Fig. 3e), confirming that the ITH defined by DEPTH is associated with genomic instability.

Tumor stage refers to the range of the primary tumor and the extent of tumor cells spread in the body. We found that DEPTH scores were significantly higher in late-stage (stage III-IV) than in early-stage (stage I-II) tumors in pan-cancer ($p = 9.09 \times 10^{-8}$) and in 5 individual cancer types (LIHC, THCA, KIRC, LUAD, and HNSC) (FDR < 0.05) (Fig. 3f). Furthermore, we compared DEPTH scores between different substages in the cancers with related data available. We found that within the same stage, DEPTH scores likely increased with the advancement of substage in pan-cancer, e.g., stage Ib > Ia, IIc > IIa or IIb, and IIIb > IIIa ($p < 0.001$) (Supplementary Fig. 1c); the same trend was observed within individual cancer types, such as Ib > Ia in ESCA, LUAD, LUSC, and CESC, IIb > IIa in CESC, and IIIb > IIIa in BRCA and UCEC ($p < 0.05$). These results indicate that DEPTH scores are likely to increase with tumor advancement. We classified tumors into two groups based on tumor size (T) and compared DEPTH scores between two groups (T1-2 versus T3-4). We found that DEPTH scores were significantly higher in T3-4 than in T1-2 tumors in pan-cancer ($p = 5.20 \times 10^{-16}$) and in 8 individual cancer types (THCA, LIHC, PRAD, CHOL, KIRC, HNSC, LUSC, and LUAD) (FDR < 0.05) (Fig. 3f). Tumor grade indicates how quickly a tumor is likely to grow and spread based on the abnormality degree of tumor cells compared to normal cells. In 9 cancer types with tumor grade information available, DEPTH scores were significantly higher in high-grade (G3-4) than in low-grade (G1-2) tumors in 4 individual cancer types (HNSC, KIRC, LIHC, and UCEC) (FDR < 0.05) (Fig. 3f). Altogether, these results indicate that the DEPTH score-based ITH increases with tumor progression, particularly in LIHC, HNSC, and KIRC.

The expression of Ki67 (encoded by the *MKI67* gene) is a marker for tumor cell proliferation[46]. We found that DEPTH scores positively correlated with *MKI67* expression levels in 13 individual cancer types ($FDR_{Sp} < 0.05$) (Fig. 3g). In the pan-cancer analysis, DEPTH scores had a significant positive correlation with *MKI67* expression levels ($p = 4.16 \times 10^{-127}$, $\rho = 0.25$). We also examined the association between two other proliferation markers (*TOP2A* and *RACGAP1*[47]) and DEPTH scores in cancer. Likewise, we observed significant positive correlations of *TOP2A* and *RACGAP1* expression levels with DEPTH scores in 13 and 15 cancer types, respectively ($FDR_{sp} < 0.05$) (Fig. 3g), as well as in pan-cancer (*TOP2A*: $p = 6.98 \times 10^{-93}$, $\rho = 0.21$; *RACGAP1*: $p = 4.14 \times 10^{-157}$, $\rho = 0.28$). Furthermore, we analyzed the correlation between DEPTH scores and a proliferation signature composed of 7 marker genes[48] and found that they had a significant positive correlation in pan-cancer and in 21 individual cancer types ($FDR_{sp} < 0.05$) (Fig. 3g). Tumor stem cell-associated characteristics ("stemness") are associated with ITH and poor outcomes in cancer[11]. Interestingly, we observed significant positive correlations between DEPTH scores and tumor stemness scores in pan-cancer ($p = 4.86 \times 10^{-52}$, $\rho = 0.16$) and in 20 individual cancer types ($FDR_{sp} < 0.05$) (Fig. 3h).

Collectively, these results indicate that the high DEPTH ITH level is likely associated with worse clinical outcomes in cancer. This is consistent with previous studies showing that ITH was an adverse prognostic factor[11].

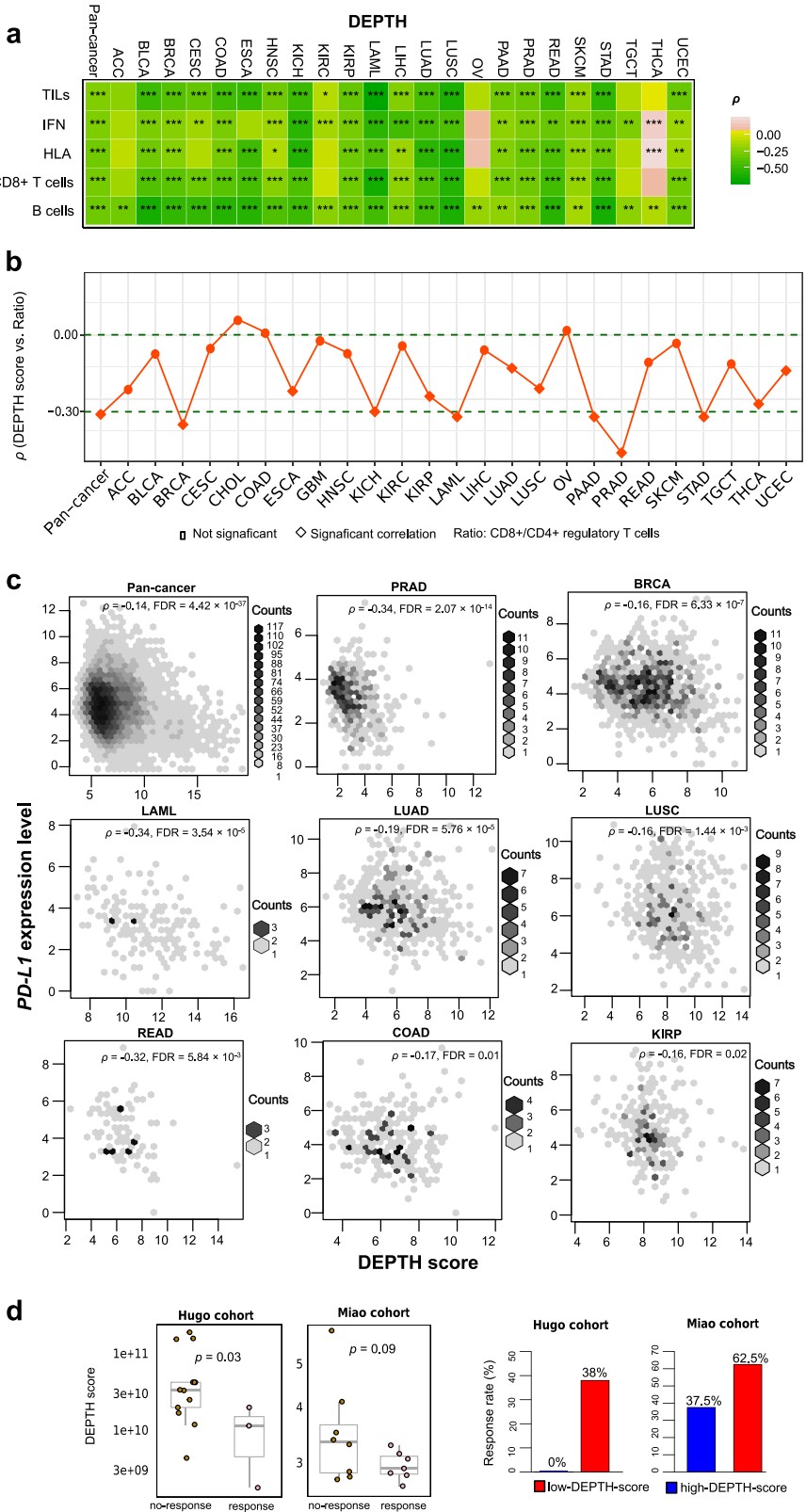

**Associations of DEPTH scores with antitumor immune response**. Many studies have shown that ITH is associated with reduced antitumor immune response[10,11,13]. We analyzed the correlations between DEPTH scores and immune signature scores in pan-cancer and within 25 individual cancer types. The immune signature scores, representing the enrichment levels of immune signatures in tumors, were the average expression levels

of all marker genes of immune signatures. A total of five antitumor immune signatures were analyzed, including B cells, CD8 + T cells, human leukocyte antigen (HLA), interferon (IFN) response, and tumor-infiltrating lymphocytes (TILs). As expected, we observed significant inverse correlations of DEPTH scores with B cell, CD8+ T cell, HLA, IFN, and TIL scores in pan-cancer (FDR < $1.0 \times 10^{-100}$, $\rho \leq -0.25$), and in 23, 18, 17, 19, and 19

**Fig. 4 Association of DEPTH scores with antitumor immunity and immunotherapy response. a** The significant inverse correlations of DEPTH scores with immune signature (B cells, CD8+ T cells, HLA, IFN, and TILs) scores in pan-cancer and in many individual cancer types (FDR$_{sp}$ < 0.05). The scores of immune signatures are the mean expression levels of their marker genes in the tumor. HLA, human leukocyte antigen. IFN, interferon. TILs, tumor-infiltrating lymphocytes. **b** The significant inverse correlations of DEPTH scores with the ratios of immune-stimulatory signature (CD8+ T cells) to immune-inhibitory signature (CD4+ regulatory T cells) in pan-cancer and in 12 individual cancer types (FDR$_{sp}$ < 0.05). The ratios are the log2-transformed values of the mean expression levels of CD8+ T cell marker genes divided by the mean expression levels of CD4+ regulatory T cell marker genes. **c** The significant inverse correlations between DEPTH scores and *PD-L1* expression levels in pan-cancer and in eight individual cancer types (FDR$_{sp}$ < 0.05). **d** The inverse correlations of DEPTH scores with the response to immunotherapy response in a melanoma cohort (Hugo cohort[51]) and a kidney cancer cohort (Miao cohort[52]) receiving the immune checkpoint blockade therapy. The DEPTH scores were compared between the responsive and non-responsive groups using the one-sided Mann–Whitney U test. The immunotherapy response rates in the high-DEPTH-score (DEPTH scores > median) tumors and in the low-DEPTH-score (DEPTH scores < median) tumors are presented.

individual cancer types, respectively (FDR$_{sp}$ < 0.05) (Fig. 4a). Moreover, DEPTH scores were inversely correlated with the ratios of immune-stimulatory signature (CD8+ T cells) to immune-inhibitory signature (CD4+ regulatory T cells) in pan-cancer ($p = 5.14 \times 10^{-181}$, $\rho = -0.31$) and in 12 individual cancer types (FDR$_{sp}$ < 0.05) (Fig. 4b). These results suggest that the high DEPTH ITH level is associated with reduced antitumor immunity, consistent with the inverse correlation between ITH and antitumor immune response[10,11,13]. Because antitumor immune signatures have a positive association with survival prognosis in cancer[49,50] and DEPTH scores were negatively associated with them, the negative correlation between DEPTH scores and survival prognosis in cancer could be a consequence of the reduced antitumor immune signatures in high-DEPTH-score tumors. To exclude this possibility, we used the Cox proportional hazards model for multivariate (DEPTH score and TILs score) survival analysis in TCGA pan-cancer and in individual cancer types. We found that the elevated DEPTH scores were negatively associated with the survival in pan-cancer (OS, DSS, DFI, and PFI) and three individual cancer types (BRCA, COAD, and KIRC) ($p < 0.02$) (Supplementary Fig. 2), consistent with the results from the previous univariate survival analysis. This suggests that the DEPTH score alone has a prognostic power.

We found that DEPTH scores were inversely associated with *PD-L1* expression levels in pan-cancer ($p = 1.7 \times 10^{-38}$, $\rho = -0.14$) and in 8 individual cancer types (FDR$_{sp}$ < 0.05) (Fig. 4c). Because both PD-L1 expression[17] and TIL infiltration levels[20] are positive predictive factors for immunotherapy response and DEPTH scores have a negative correlation with both of them, we expected that the high DEPTH scores would be associated with the reduced response to immunotherapy. As expected, in a melanoma cohort (Hugo cohort[51]) and a KIRC cohort (Miao cohort[52]) receiving the ICB immunotherapy, the responsive group had significantly lower DEPTH scores than the non-responsive group ($p = 0.03$, $0.09$ for Hugo and Miao cohorts, respectively) (Fig. 4d). Furthermore, on the basis of DEPTH scores, we divided patients into two groups (high-DEPTH-score (DEPTH scores > median) versus low-DEPTH-score (DEPTH scores < median)) and compared the ICB response rate between both groups. We found that the high-DEPTH-score group had a lower response rate than the low-DEPTH-score group (0% versus 38% in the Hugo cohort; 37.5% versus 62.5% in the Miao cohort) (Fig. 4d). These results suggest that the high DEPTH ITH level is likely to be associated with the reduced response to immunotherapy in cancer.

**Association of DEPTH scores with drug response in cancer.** Based on the data from the Genomics of Drug Sensitivity in Cancer (GDSC) project (https://www.cancerrxgene.org), we found that DEPTH scores had significant correlations with drug sensitivity (IC50 values) to 180 (68%) of 265 compounds tested in cancer cell lines (FDR$_{sp}$ < 0.05) (Supplementary Data 1). Among

the 180 compounds, 144 displayed a significant negative correlation of IC50 values with DEPTH scores versus 36 showing a significant positive correlation in the cancer cell lines (Fig. 5a and Supplementary Data 1). These data suggest that the DEPTH score-based ITH is associated with the sensitivity to a broad spectrum of anticancer drugs. Interestingly, we found many compounds to which the increased sensitivity was associated with the high ITH (Supplementary Data 1). A possible explanation is that the elevated expression of oncogenic signatures in high-ITH tumors enhances their sensitivity to relevant inhibitors. Furthermore, we analyzed the correlation between DEPTH scores and drug response in cancer patients using TCGA, TARGET, and three GEO (GSE1379, GSE107850, and GSE123728) datasets. In TCGA, because different types of cancer patients were treated with different drugs and the drug usage data were incomplete in many individual cancer types, we performed the analysis only in pan-cancer. When all drugs were taken into account, the responsive group had significantly lower DEPTH scores than the non-responsive group ($p = 0.0001$) (Fig. 5b). In TARGET, we found that acute myeloid leukemia (AML) patients with minimal residual disease (MRD) at the end of the second course of induction therapy had significantly higher DEPTH scores than those without MRD ($p = 0.059$) (Fig. 5c). Moreover, in the acute lymphoblastic leukemia (ALL) phase II patients, DEPTH scores displayed a significant positive correlation with the MRD percentages after 8 days of remission induction therapy ($p = 0.012$, $\rho = 0.25$) (Fig. 5c). In GSE1379[53], the breast cancer patients responsive to tamoxifen had lower DEPTH scores than the patients not responsive to tamoxifen ($p = 0.03$) (Fig. 5d). In GSE107850[54], the lower-grade glioma patients who were responsive to the combination therapy of chemotherapy (temozolomide) and radiotherapy had lower DEPTH scores than the patients who were not responsive to such therapy ($p = 0.0007$) (Fig. 5d). In GSE123728[55], the melanoma patients with favorable response to pembrolizumab had lower DEPTH scores than the patients with an unfavorable response to the drug ($p = 0.088$) (Fig. 5d). Overall, these data indicate that DEPTH scores likely have a significant association with drug response in cancer.

**Molecular features associated with DEPTH scores.** We identified 262 genes whose expression alteration had a strong positive correlation with DEPTH scores in at least five cancer types (FDR < 0.05, $\rho > 0.5$) (Fig. 6a and Supplementary Data 2). Notably, many of these genes encode human cytokines and growth factors, including *CCL14*, *CCL21*, *CTSG*, *CXCL12*, *FGF10*, *FGF7*, *GDF10*, *GREM2*, *IL33*, *KL*, *OGN*, *PDGFD*, and *PDGFRA*. The essential roles of growth factors and cytokines in driving cancer cell proliferation and invasion have been well recognized[56,57]. The 262 genes also included many marker genes of human leukocyte and stromal cell differentiation, including *ABCB1*, *ACKR1*, *CD1C*, *CD34*, *CLEC10A*, *GYPC*, *JAM2*, *KIT*, *PDGFRA*, *SELP*, *SIGLEC6*, and *TEK*. A considerable number of transcription factor genes

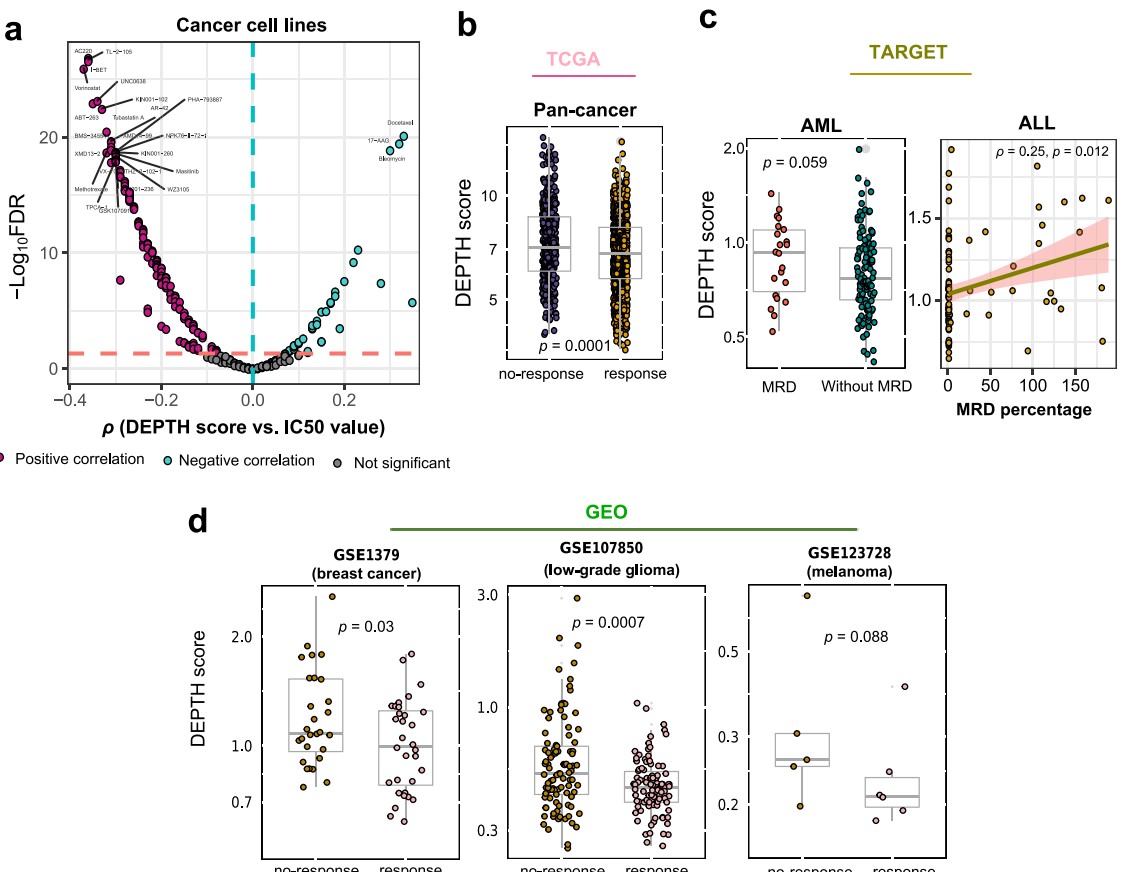

**Fig. 5 Association of DEPTH scores with drug response in cancer. a** Correlations between DEPTH scores and drug sensitivity (IC50 values) of 265 compounds in cancer cell lines. The names of compounds with |ρ| > 0.3 are shown. **b** Comparison of DEPTH scores between the drug-responsive group and the non-responsive group in TCGA pan-cancer. **c** Comparison of DEPTH scores between the AML patients with MRD at the end of the second course of induction therapy and those without MRD, and the correlation between DEPTH scores and the MRD percentages after 8 days of remission induction therapy in the ALL phase II patients, using the TARGET datasets. AML, acute myeloid leukemia. ALL, acute lymphoblastic leukemia. MRD, minimal residual disease. **d** DEPTH scores are significantly higher in the cancer patients with better drug response than in those with worse drug response, as shown in three GEO datasets (GSE1379[53], GSE107850[54], and GSE123728[55]).

were also in the 262-gene list, including *AFF3, CBX7, FHL1, FHL5, FOXF1, HLF, LDB2, LMO3, MEF2C, MEOX1, MEOX2, NR2F1, PGR, PKNOX2, RUNX1T1, TCF21, ZBTB16, ZEB1,* and *ZNF208.* Pathway analysis by GSEA[34] revealed that these genes were mainly involved in pathways of calcium signaling, cell adhesion molecules, ABC transporters, focal adhesion, regulation of actin cytoskeleton, chemokine signaling, leukocyte transendothelial migration, pathways in cancer, tight junction, MAPK signaling, ECM-receptor interaction, tyrosine metabolism, and gap junction (Fig. 6a). These results indicate that the expression alterations in the genes involved in cell growth and proliferation, tumor-stroma crosstalk, and immune signatures may contribute significantly to the DEPTH score-based ITH.

We found 15 genes whose somatic mutations were associated with increased DEPTH scores in at least five cancer types (FDR < 0.1). These genes included *AHNAK2, TP53, PLEC, COL5A1, FAT3, PCDH17, RIMS1, ASH1L, CDH23, DNAH17, DYNC2H1, MYCBP2, SLIT3, TRRAP,* and *USH2A.* Interestingly, we found 9 of the 15 genes whose mutations were significantly associated with worse OS in pan-cancer (log-rank test, *p* < 0.05) (Supplementary Fig. 3). This is consistent with previous observations that the high DEPTH ITH level was associated with unfavorable clinical outcomes in cancer. Notably, many of these genes were involved in the calcium signaling pathway, including *AHNAK2, PLEC, COL5A1, FAT3, PCDH17, RIMS1, CDH23,* and *SLIT3.*

Besides, several genes, including *TP53* and *TRRAP*, were involved in DDR regulation to maintain genomic stability.

We identified 40 proteins whose expression levels were significantly higher in high-DEPTH-score than in low-DEPTH-score tumors in at least five cancer types (two-sided Student's *t* test, FDR < 0.05) (Supplementary Data 3). These proteins were mainly involved in cell cycle regulation (such as Cyclin_B1, Cyclin_E1, Cyclin_E2, FoxM1, and Chk2), DDR (such as PCNA, MSH6, MSH2, XRCC1, BAP1, and Ku80), and metabolism (such as GAPDH, ACC1, TIGAR, and FASN). Also, there were 33 proteins whose expression levels were significantly lower in high-DEPTH-score than in low-DEPTH-score tumors in at least five cancer types (two-sided Student's *t* test, FDR < 0.05) (Supplementary Data 3). Some of these proteins were tumor suppressors, including Rb, FOXO3, and p27, and many proteins were kinases and involved in the regulation of cell proliferation, differentiation, and apoptosis, such as c-KIT, AKT, EGFR, MAPK1/3, PRKCA, SRC, ACVRL1, and JNK2.

GSEA[34] identified a number of KEGG pathways that were highly enriched in high-DEPTH-score tumors and low-DEPTH-score tumors (FDR < 0.05). The pathways highly enriched in high-DEPTH-score tumors in at least five cancer types mainly associated with DNA damage response and cell cycle, consistent with previous results that the upregulated proteins in high-DEPTH-score tumors were associated with cell cycle regulation and DDR.

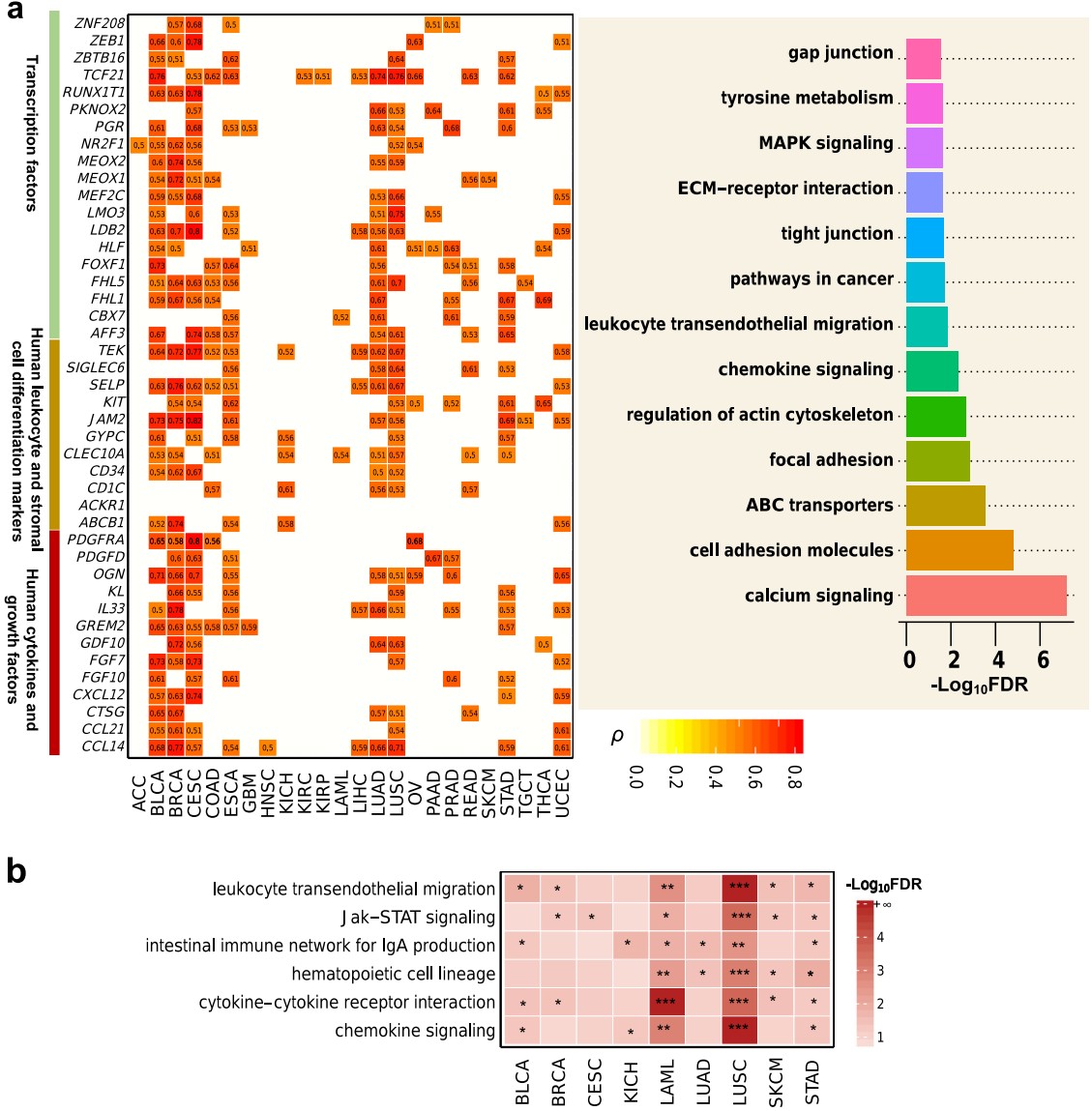

**Fig. 6 Genes and pathways whose expression alteration or enrichment levels are associated with DEPTH scores in cancer. a** Genes whose expression alteration is strongly associated with DEPTH scores in at least five cancer types (FDR < 0.05, ρ > 0.5), and their associated pathways identified by GSEA[34] (FDR < 0.05). **b** The immune-related pathways highly enriched in low-DEPTH-score versus high-DEPTH-score tumors in at least five cancer types identified by GSEA[34] (FDR < 0.05).

The pathways highly enriched in low-DEPTH-score tumors were mainly immune relevant, including cytokine–cytokine receptor interaction, Jak–STAT signaling, leukocyte transendothelial migration, chemokine signaling, hematopoietic cell lineage, and intestinal immune network for IgA production (Fig. 6b). Again, these results indicate the strong inverse association between DEPTH scores and antitumor immune signatures.

**DEPTH scores across and within individual cancer types.** We used the Shannon diversity index (SDI) to define the DEPTH score diversity in each cancer type, where a high index value indicates a highly diverse ITH distribution and a low value indicates a homogenous ITH distribution in a cancer type. Among the 25 cancer types, PAAD, CHOL, SKCM, GBM, and LIHC had the highest SDI values, while OV, KICH, ESCA, THCA, and UCEC had the lowest SDI values (Fig. 7a). The SDI score reflects the variation of DEPTH scores within a cancer type in that they had a strong positive correlation across the 25 cancer types ($\rho = 0.96$) (Fig. 7a). The SDI score also indicates the

intertumor heterogeneity across the tumor samples in the same cancer type. We found that the SDI scores displayed a negative correlation with the median tumor purity ($\rho = -0.42$) and a positive correlation with the median ploidy scores across the 25 cancer types ($\rho = 0.41$) (Fig. 7a). Moreover, the cancer types with high SDI scores (> median) had a higher median TMB than the cancer types with low SDI scores (< median) ($p = 0.06$) (Fig. 7b). These results indicate that the diversity of DEPTH score-based ITH distribution is positively associated with genomic instability, while it is negatively associated with tumor purity. Furthermore, like certain DNA-based ITH evaluation methods[5,58], we calculated the number of clones in each tumor sample based on DEPTH scores. We defined the clone number in a tumor sample as the ratio (rounded number) of its DEPTH score to the smallest DEPTH score in pan-cancer. Within pan-cancer, the inferred clone numbers ranged between 1 and 24, with 98% of the tumors encompassing at least 3 clones and 80% having less than 10 clones (Fig. 7c and Supplementary Data 4). We correlated the clone numbers inferred by DEPTH with those inferred by two

DNA-based methods[5,58] in pan-cancer and found that they had significant correlations ($p < 0.001$, $\rho = 0.34$, 0.13, respectively) (Fig. 7d). Within individual cancer types, SKCM and TGCT had relatively high clone numbers, while PRAD and THCA had low clone numbers (Fig. 7e), consistent with the results inferred by EXPANDS[4,5]. The clone numbers in SKCM ranged between 7 and 23 (median = 19), with 98% of tumors encompassing at least 10 clones; the clone numbers in TGCT ranged between 8 and 24 (median = 20), with 99% of tumors encompassing more than 10 clones. In contrast, the clone numbers in PRAD ranged between 1 and 14 (median = 3), with 93% of tumors having no more than 5 clones; the clone numbers in THCA ranged between 1 and 10 (median = 5), with 68% of tumors having no more than 5 clones. Interestingly, SKCM had high SDI and high clone numbers, suggesting that this type of cancer is characterized by high intertumor and ITH. In contrast, THCA displayed low SDI and low clone numbers, suggesting its low intertumor and ITH. The inferior tumor heterogeneity may explain why THCA is generally unaggressive and has an excellent prognosis[59].

**Association of DEPTH scores with tumor purity**. We found that high DEPTH scores were significantly associated with increased tumor purity in 15 cancer types ($FDR_{sp} < 0.05$) (Fig. 8a). This is in line with the finding that high DEPTH scores were associated with reduced TIL levels. Meanwhile, this indicates that the ITH defined by DEPTH is more prominent in tumor cells than in non-tumor cells. To correct the effect of tumor purity on the associations of DEPTH scores with immune signatures and genomic instability, we used logistic regression with two predictors (DEPTH score and tumor purity) to predict the five antitumor immune signature scores and TMB in pan-cancer and 25 individual cancer types. We observed that the DEPTH score was a significant negative predictor for the antitumor immune signatures in most cases (Fig. 8b and Supplementary Fig. 4). Meanwhile, the DEPTH score was a positive predictor for TMB in most cases, where the β value was greater than 1 in 10 individual cancer types (Fig. 8c). In pan-cancer, DEPTH score was a significant positive predictor for TMB (TMB: $\beta = 0.63$, $p = 2.66 \times 10^{-13}$). Also, DEPTH scores still displayed a significant inverse correlation with survival prognosis in pan-cancer and multiple individual cancer types after correcting tumor purity (Fig. 8d). These results suggest that DEPTH scores are prominently associated with immune signatures, genomic instability, and survival prognosis in cancer, regardless of tumor purity. To further demonstrate that the DEPTH score is an authentic measure of ITH, we calculated the DEPTH scores in the TCGA and GTEx normal tissue whose tumor purity is supposed to be zero. As expected, in pan-cancer and 25 individual cancer types, the tumors had much higher DEPTH scores than normal tissue ($p < 0.05$) (Fig. 8e).

**Relationship between different ITH measures**. We explored the correlation between ITH scores defined by seven different methods (DEPTH, tITH[24], sITH[26], PhyloWGS[6], EXPANDS[4,5], ABSOLUTE[2], and MATH[3]) in TCGA pan-cancer and individual cancer types. As expected, 19 of the 21 pairwise correlations were significantly positive in pan-cancer ($p < 0.01$, $0.09 \le \rho \le 0.55$) (Fig. 9). The strongest correlation was observed between DEPTH and tITH ($\rho = 0.55$), and the next was between DEPTH and sITH ($\rho = 0.50$), suggesting that DEPTH scores have a stronger correlation with the other mRNA-based ITH scores than with the DNA-based ITH scores. Surprisingly, the correlation between DEPTH and sITH was stronger than that between tITH and sITH ($\rho = 0.30$), although both tITH and sITH were developed based on the Jensen-Shannon Divergence measure[24,26]. Among the

DNA-based methods, PhyloWGS and ABSOLUTE had the strongest correlation ($\rho = 0.30$), and the correlations of PhyloWGS with EXPANDS and MATH were also stronger than those between EXPANDS, ABSOLUTE, and MATH. A potential reason behind this is that PhyloWGS evaluates the ITH based on both mutations and CNAs in tumor cells to make it correlative with both mutations-based (EXPANDS and MATH) and CNAs-based (ABSOLUTE) methods.

Within individual cancer types, the correlations between ITH scores derived from the different methods were mostly positive, and few were negative (Supplementary Data 5). DEPTH scores were positively correlated with ITH scores from tITH, sITH, PhyloWGS, EXPANDS, MATH, and ABSOLUTE in 25, 15, 6, 3, 4, and 6 cancer types, respectively ($FDR_{sp} < 0.05$). DEPTH showed the strongest correlation with tITH in almost all 25 cancer types ($0.53 \le \rho \le 0.90$) and showed a strong correlation with sITH in 15 cancer types ($0.28 \le \rho \le 0.71$). The correlation between tITH and sITH was significant in 9 cancer types ($0.16 \le \rho \le 0.61$) but was likely to be weaker than that between DEPTH and sITH in almost all individual cancer types analyzed (Supplementary Data 5). The other correlations significant in more than 10 cancer types were those between ITH scores from the DNA-based methods, including MATH versus ABSOLUTE, MATH versus PhyloWGS, and ABSOLUTE versus PhyloWGS. Overall, these results indicate that the ITH measures within the DNA or mRNA level are more correlated than those between DNA and mRNA levels.

**Validation of DEPTH scores in other datasets**. To demonstrate the DEPTH algorithm's reliability and robustness, we analyzed the correlations of DEPTH scores with clinical features and immune signatures in 42 other gene expression profiling datasets generated by different technologies and platforms and containing different numbers of genes. In 37, 33, 37, 27, 34 datasets, DEPTH scores showed significant positive correlations with stemness scores, *MKI67*, *TOP2A*, and *RACGAP1* expression levels, and proliferation signature scores, respectively (Spearman's correlation test, $p < 0.05$) (Fig. 10a). In 22 datasets, DEPTH scores had significant inverse correlations with at least three of the five immune signatures, and in 17 datasets, DEPTH scores had significant inverse correlations with the ratios of CD8+/CD4+ regulatory T cells (Spearman's correlation test, $p < 0.05$) (Fig. 10b). In three large-scale datasets, including breast cancer datasets E-MTAB-6703 (sample size $n = 2302$) and METABRIC ($n = 1980$), and gastric cancer dataset ACRG ($n = 300$), the high-DEPTH-score tumors showed significantly worse OS than the low-DEPTH-score tumors (log-rank test, $p < 0.05$) (Fig. 10c). In addition, in 11 datasets involving the tumor grade phenotype data, DEPTH scores were remarkably higher in high-grade than in low-grade tumors in 7 datasets ($p < 1.0 \times 10^{-4}$) (Fig. 10d).

In TARGET, the AML patients with certain cytogenetic abnormalities, such as t(9;11)(p22;q23), t(10;11)(p11.2;q23), del5q, trisomy 21, and *MLL1* translocation, had significantly higher DEPTH scores than the AML patients without such abnormalities ($p < 0.05$) (Fig. 10e), confirming the positive correlation between DEPTH scores and chromosomal/genomic instability. Moreover, DEPTH scores had a significant positive correlation with the bone marrow leukemic blast percentages of AML patients ($p = 0.003$, $\rho = 0.27$) (Fig. 10f). DEPTH scores were higher in the AML patients at the undifferentiated stage (M0) than in those at the differentiated stage (M1-7) and were higher in the AML patients with other sites of relapse than in those without other sites of relapse ($p < 0.05$) (Fig. 10f). Furthermore, the high-risk AML patients showed higher DEPTH scores than the low-risk AML patients, and we observed a similar result in neuroblastoma (NBL)

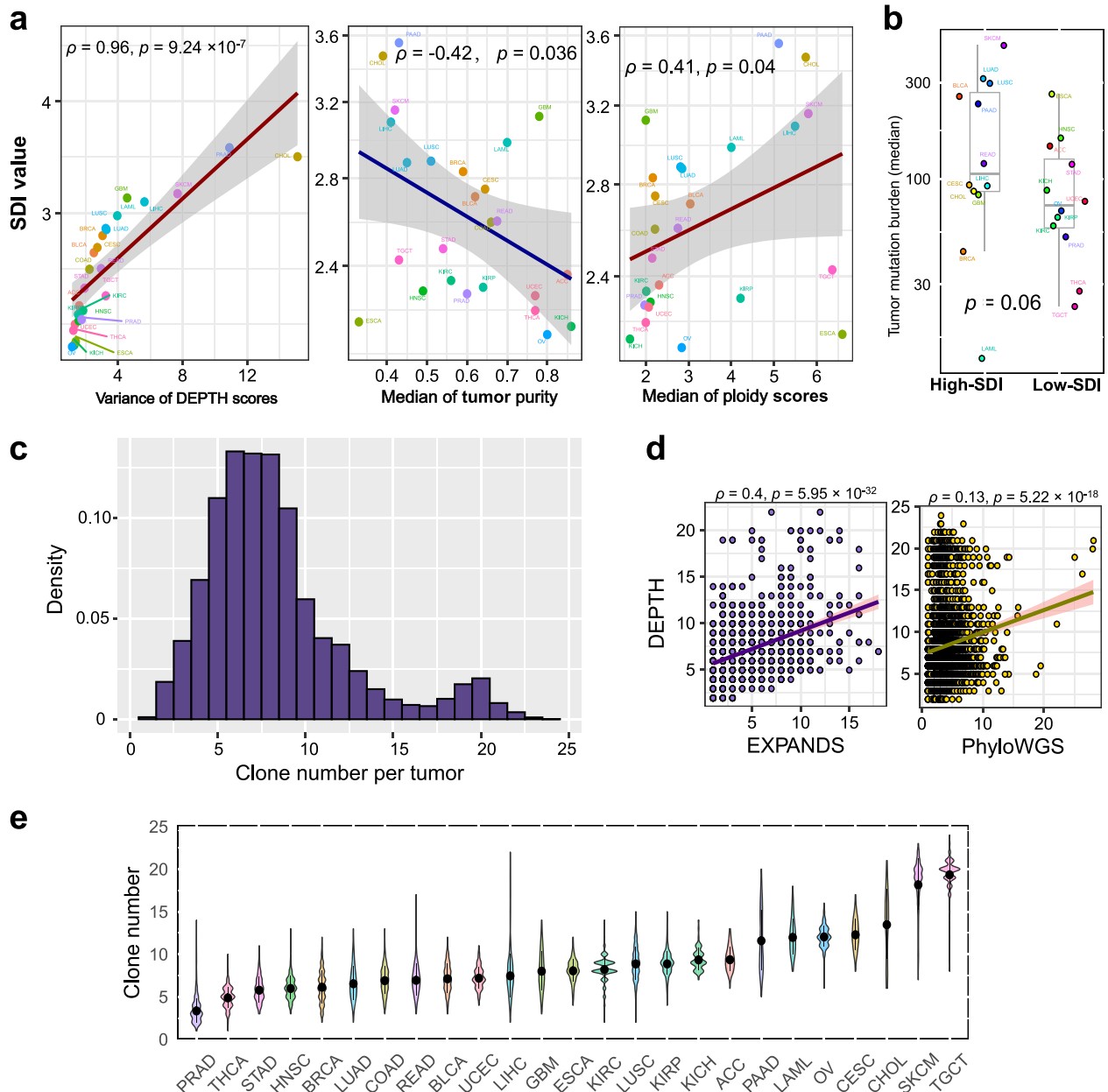

**Fig. 7 Comparison of the DEPTH score-based ITH across and within individual cancer types. a** The Shannon diversity index (SDI) values and their association with the variation of DEPTH scores, the median of tumor purity, and median tumor ploidy scores in 25 cancer types. **b** The cancer types with high SDI scores (> median) show a higher median TMB than the cancer types with low SDI scores (< median). **c** Clone number distribution inferred by DEPTH across cancer types. **d** Correlations between the clone numbers inferred by DEPTH and those inferred by EXPANDS[4, 5] and PhyloWGS[6] in pan-cancer. **e** Clone number distribution inferred by DEPTH within individual cancer types.

patients ($p = 0.04$, 0.003 for AML and NBL, respectively) (Fig. 10g). The pan-cancer analysis showed that the high DEPTH scores were associated with worse OS (log-rank test, $p = 0.014$) (Fig. 10h). These results confirmed the positive correlation between DEPTH scores and clinical risk in cancer.

Overall, these results obtained from validation datasets were consistent with the findings in TCGA datasets, demonstrating the reliability and robustness of the DEPTH algorithm in evaluating the ITH level.

## Discussion
ITH is associated with tumor progression, immune evasion, and drug resistance. Therefore, the ITH level is a biomarker of cancer

prognosis and therapy response. Most of the previous studies evaluating ITH levels were based on DNA alterations, and a few were based on mRNA alterations. In this study, we proposed the DEPTH algorithm, a new mRNA-based method for evaluating ITH. We demonstrated that high DEPTH scores have a prevalent association with the common features of high-ITH tumors, e.g., genomic instability, tumor advancement, unfavorable prognosis, immunosuppression, and drug resistance. We compared DEPTH with the other six methods for evaluating ITH and found that DEPTH scores tended to display a stronger and more consistent correlation with these features than the ITH scores derived from the other methods in TCGA pan-cancer and 25 individual cancer types (Supplementary Note 1). Notably, compared to two mRNA-based methods[24,26], particularly the gene expression profiling

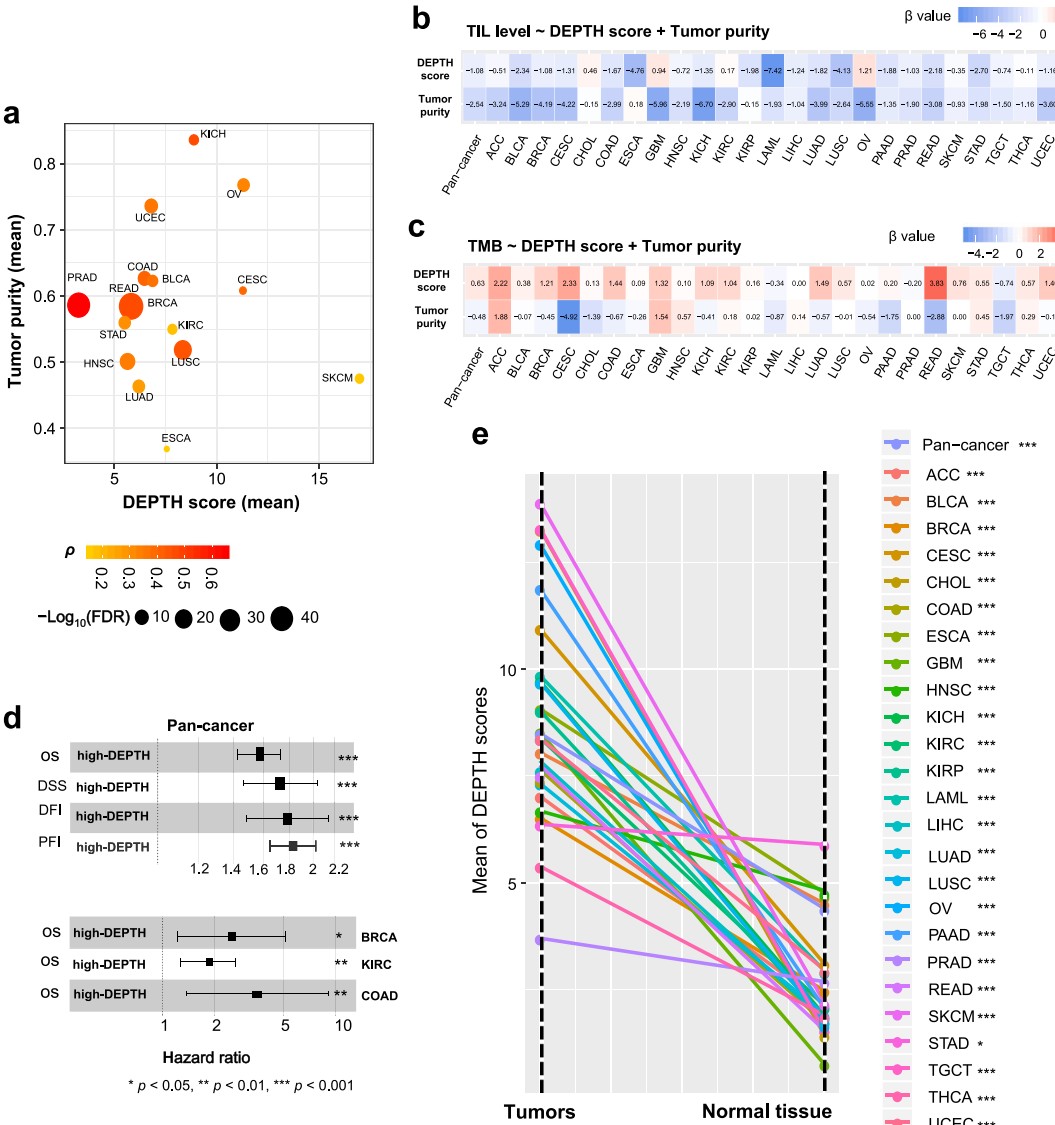

**Fig. 8 Association of DEPTH scores with tumor purity. a** The positive correlation between DEPTH scores and tumor purity in 15 cancer types (FDR$_{sp}$ < 0.05). **b, c** Logistic regression analysis showing that DEPTH scores have a significant negative and a significant positive correlation with antitumor immune signatures (**b**) and TMB (**c**) in most cases after correcting tumor purity. **d** Cox proportional hazards regression analysis showing that DEPTH scores have a significant inverse correlation with survival prognosis in pan-cancer and in multiple individual cancer types after correcting tumor purity. The hazard ratio for the high DEPTH score (upper third) was calculated relative to the hazard for the individuals in the low-DEPTH score (bottom third), and its 95% CI is shown as whiskers. **e** Comparison of DEPTH scores between tumors and normal tissue.

alterations-based method tITH[24], DEPTH exhibits certain superiorities in quantifying ITH. First, DEPTH scores show stronger associations with genomic instable features, such as high TMB, *TP53* mutations, and HRD. Second, high DEPTH scores have more significant associations with unfavorable clinical features, including poor survival, tumor advancement, strong tumor cell proliferation potential, and high tumor stemness. Finally, high DEPTH scores have a stronger and more consistent association with drug response.

In general, the correlation between ITH scores from the same molecule type (mRNA versus mRNA; DNA versus DNA) is stronger than that between different molecule types (mRNA versus DNA), suggesting the molecule type-specific ITH. The two transcriptome-based ITH scores (DEPTH and tITH) display prominently stronger negative correlations with antitumor immune signatures than the DNA-based ITH scores. In addition, in multiple cancer types, DEPTH and tITH scores have a positive correlation with tumor purity, as compared to the negative correlation of PhyloWGS ABSOLUTE, and MATH scores with tumor purity, again demonstrating the distinction between the mRNA- and DNA-based ITH.

correlation with MSI, as compared to the negative correlation of MATH and ABSOLUTE scores with MSI, suggesting the molecule type-dependent correlation between ITH and MSI. However, with the other genomic instable features (TMB, *TP53* mutations, and HRD), both mRNA- and DNA-based ITH scores are likely to be positively correlated. Furthermore, in multiple cancer types, DEPTH and tITH scores have a positive correlation with tumor purity, as compared to the negative correlation of PhyloWGS ABSOLUTE, and MATH scores with tumor purity, again demonstrating the distinction between the mRNA- and DNA-based ITH.

Compared to the DNA-based ITH scores, DEPTH scores show a much stronger correlation with the mRNA-based measures, including immune signatures, cell proliferation, and stemness. Moreover, DEPTH scores have a stronger correlation with tumor progression and prognosis phenotypes (tumor size, stage, grade, survival, and drug response) than the DNA-based ITH scores.

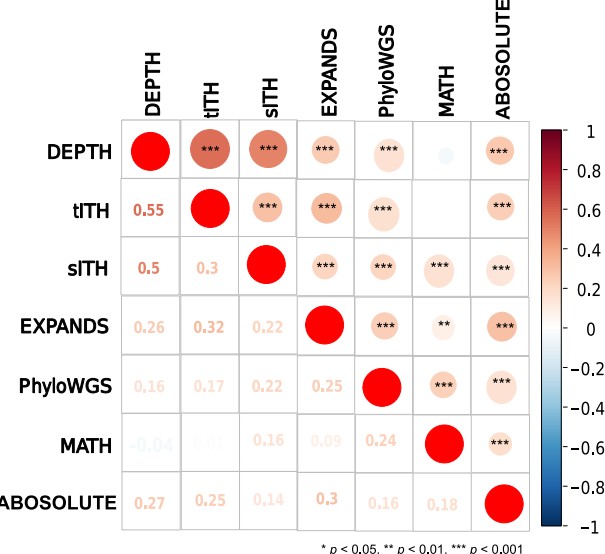

**Fig. 9 Relationship between different ITH scores.** The pairwise correlations between ITH scores inferred by seven different methods (DEPTH, tITH[24], sITH[26], PhyloWGS[6], EXPANDS[4,5], ABSOLUTE[2], and MATH[3]) in pan-cancer are shown.

Furthermore, DEPTH scores even display a stronger correlation with specific DNA-based measures, such as TMB, than most of the DNA-based ITH scores (Supplementary Note 1). DEPTH scores can capture genetic alterations in cancer to a higher degree than tITH in that DEPTH scores show a stronger correlation with the genetic measures (TMB, *TP53* mutations, and HRD).

Within the individual cancer types, the different ITH measures exhibited certain commonality and distinction. For example, among the 25 cancer types, DEPTH scores were relatively high in SKCM and TGCT and low in PRAD and THCA (Supplementary Fig. 5). A similar trend was observed in the ITH scores from tITH, ABSOLUTE, and EXPANDS (Supplementary Fig. 6). In phyloWGS, although the ITH scores were also low in PRAD and THCA, they were intermediate in SKCM and TGCT. In sITH, the ITH scores were low in PRAD and THCA, but even lower in SKCM. Surprisingly, for the somatic mutation profiles-based ITH evaluation method MATH, the ITH scores were close between SKCM, TGCT, PRAD, and THCA, although SKCM has higher TMB than PRAD and THCA[60]. In BRCA, LUAD, and KIRC, the mRNA-based rather than DNA-based ITH scores showed a prognostic power. In contrast, in ACC, PAAD, ESCA, and HNSC, the DNA-based instead of mRNA-based ITH scores exhibited a prognostic power. In COAD and UCEC, both mRNA- and DNA-based ITH scores were prognostic. In addition, in HNSC, LIHC, LUAD, LUSC, and THCA, the mRNA-based ITH scores were more likely to be associated with tumor progression phenotypes (tumor size, stage, and grade) than the DNA-based ITH scores, while in UCEC and BLCA, the DNA-based ITH scores were more associated with them. In KIRC, both mRNA- and DNA-based ITH scores associated with tumor progression phenotypes. These data may suggest complementarity between the DNA- and mRNA-based methods for evaluating ITH.

One prominent advantage of DEPTH over the other ITH evaluation algorithms is that DEPTH also applies to the situation of gene expression profiles in normal samples being unavailable. In this case, we calculate the DEPTH score of a tumor sample based on the variation of gene expression values in the tumor sample from mean gene expression values of all tumor samples. Using this alternative method, we recalculated DEPTH scores in the 25 TCGA cancer types. We found that the new DEPTH scores

exhibited similar characteristics with the previous DEPTH scores, such as their significant association with genome instability, negative association with clinical outcomes (survival prognosis, tumor progression, cell proliferation, and stemness), negative association with antitumor immune signature scores, and positive association with tumor purity (Supplementary Data 7). These results suggest that the new DEPTH scores are a viable alternative for measuring ITH when gene expression profiles in normal samples are not available.

In conclusion, the DEPTH algorithm is superior to or comparable with the other algorithms in evaluating ITH and may provide new insights into tumor biology and potential clinical implications for cancer prognosis and treatment.

## Methods

**Datasets.** We downloaded cancer genomics datasets for 25 TCGA cancer types from the genomic data commons data portal (https://portal.gdc.cancer.gov/). For each cancer type, we obtained its gene expression profiles (RNA-Seq V2, level 3 and RSEM normalized), somatic mutation profiles (level 3), protein expression profiles (level 3), and clinical data. Because seven (ACC, CESC, LAML, OV, PAAD, SKCM, and TGCT) TCGA RNA-Seq datasets contained no normal control samples, we used the combined TCGA and GTEx gene expression (RSEM normalized) data from the UCSC Xena project (https://xenabrowser.net/datapages/). Before analysis, all RSEM normalized gene expression values were added 1 and then log2-transformed. We downloaded validation datasets from four resources: GEO (https://www.ncbi.nlm.nih.gov/geo/), ArrayExpress (https://www.ebi.ac.uk/arrayexpress/), TARGET (https://ocg.cancer.gov/programs/target), and UCSC (https://xenabrowser.net/datapages/). The cancer cell line datasets for gene expression profiles (Affymetrix Human Genome U219 array) and drug sensitivities (IC50 values) were downloaded from the Genomics of Drug Sensitivity in Cancer (GDSC) project (https://www.cancerrxgene.org/downloads). We also downloaded several datasets that contained gene expression profiling and clinical drug response data in cancer from TARGET and GEO. A summary of these datasets is presented in Supplementary Data 8. The marker gene sets representing different immune signatures were from several publications, including B cells[61], CD8+ T cells[61], CD4 + regulatory T cells[61], HLA[50], IFN[61], and TILs[62]. Seven proliferation signature marker genes were obtained from the publication[48], and 109 "stemness" gene signatures were from the publication[11]. These gene sets are listed in Supplementary Data 9.

**Algorithm.** Given a normalized gene expression profiling dataset containing $m$ genes and $s$ samples ($t$ tumor samples and $n$ normal controls), the heterogeneity level (DEPTH score) of tumor sample $TS$ is defined as

$$\sqrt{\frac{1}{m-1}\left(\sum_{i=1}^{m}\left(\left(ex(G_i,TS)-\frac{1}{n}\sum_{j=1}^{n}ex(G_i,NS_j)\right)^2-\frac{1}{m}\sum_{i=1}^{m}\left(ex(G_i,TS)-\frac{1}{n}\sum_{j=1}^{n}ex(G_i,NS_j)\right)^2\right)^2\right)}$$

$$=\sqrt{\frac{1}{m-1}\left(\sum_{i=1}^{m}\left(\Delta(G_i,TS,NS_j)^2-\frac{1}{m}\sum_{i=1}^{m}\Delta(G_i,TS,NS_j)^2\right)^2\right)},\ \Delta(G_i,TS,NS_j)$$

$$=ex(G_i,TS)-\frac{1}{n}\sum_{j=1}^{n}ex(G_i,NS_j),$$

where $ex(G_i, TS)$ denotes gene $G_i$ expression level in $TS$ and $ex(G_i, NS_j)$ $G_i$ expression level in normal sample $NS_j$. If normal controls are not available in the gene expression profiling dataset, the DEPTH score of tumor sample $TS$ is defined as

$$\sqrt{\frac{1}{m-1}\left(\sum_{i=1}^{m}\left(\left(ex(G_i,TS)-\frac{1}{t}\sum_{j=1}^{t}ex(G_i,CS_j)\right)^2-\frac{1}{m}\sum_{i=1}^{m}\left(ex(G_i,TS)-\frac{1}{t}\sum_{j=1}^{t}ex(G_i,CS_j)\right)^2\right)^2\right)}$$

$$=\sqrt{\frac{1}{m-1}\left(\sum_{i=1}^{m}\left(\Delta(G_i,TS,CS_j)^2-\frac{1}{m}\sum_{i=1}^{m}\Delta(G_i,TS,CS_j)^2\right)^2\right)},\ \Delta(G_i,TS,CS_j)$$

$$=ex(G_i,TS)-\frac{1}{t}\sum_{j=1}^{t}ex(G_i,CS_j),$$

where $ex(G_i, CS_j)$ denotes $G_i$ expression level in tumor sample $CS_j$.

DEPTH calculated ITH scores based on the standard deviations of the gene expression values variations in tumors from mean gene expression values in normal or tumor samples for more than 1000 genes. The DEPTH algorithm (R function) is available at https://github.com/WangX-Lab/DEPTH/ under a GNU GPL open-source license.

**Definition of the DEPTH score diversity.** To determine the DEPTH score diversity (intertumor heterogeneity) of an individual cancer type, we used the

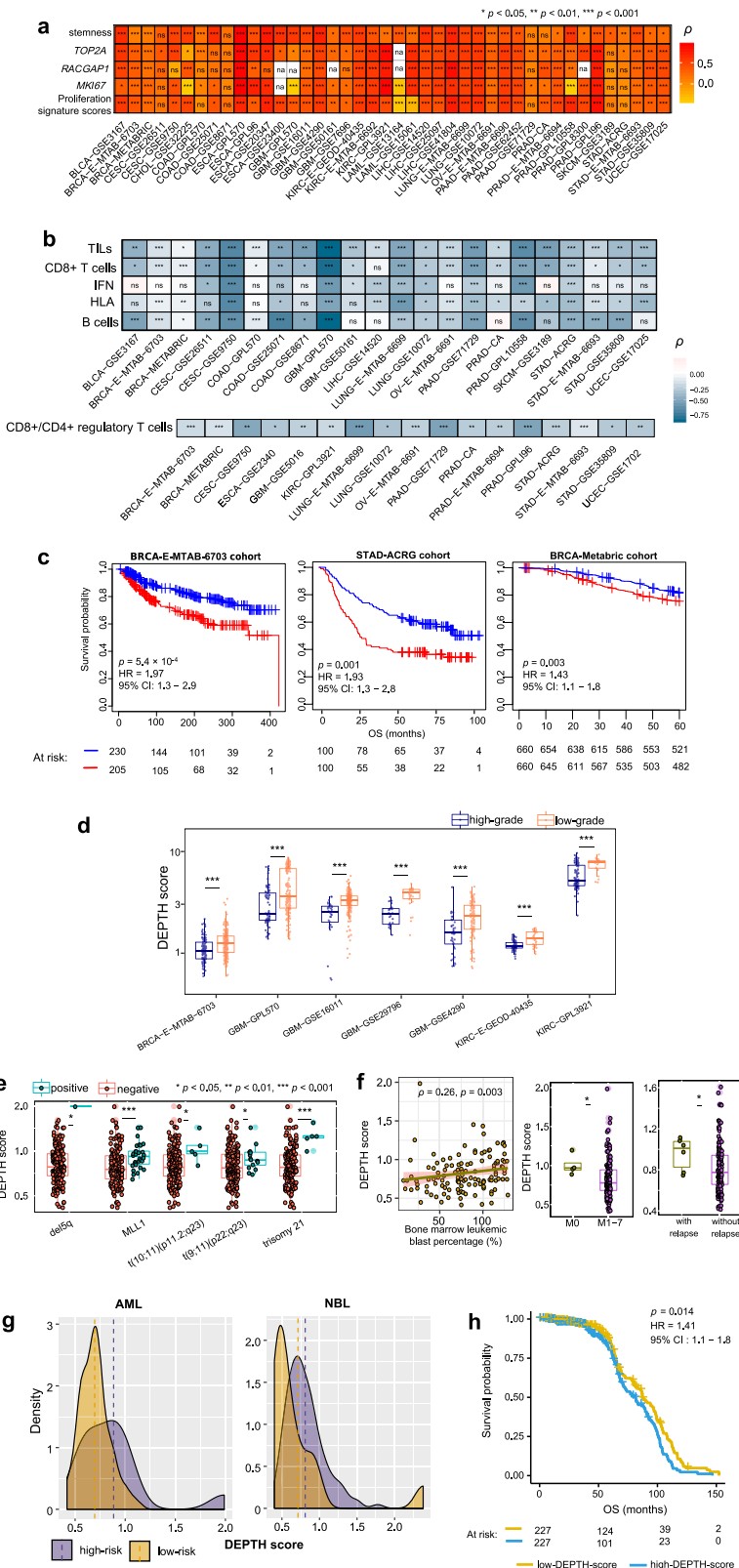

Shannon diversity index:

$$\mathrm{SDI} = -\sum_{i=1}^{n} p_i \log_2 p_i,$$

where $n$ is the number of intervals dividing the DEPTH scores of all tumor samples in the cancer type, and $p_i$ is the proportion of tumor samples whose DEPTH scores lie in the $i$th interval. Here we set $n$ as 22 and the $i$th interval as $[i, i+1)$.

**ITH scores by other methods.** MATH scores were calculated using the function "math.score"[3] in R package "maftools" with the input of "maf" files, which were obtained from the genomic data commons data portal (https://portal.gdc.cancer. gov/). Ploidy scores and tumor purity were calculated using ABSOLUTE[2] with the input of "SNP6" files, which were obtained from the genomic data commons data portal (https://portal.gdc.cancer.gov/). From the associated publications[5,26,58], we obtained the ITH scores or clone numbers inferred by EXPANDS, PhyloWGS, and sITH in 11, 24, and 19 TCGA cancer types, respectively. For the ITH scores defined

**Fig. 10 Association of DEPTH scores with clinical features and immune signatures in validation datasets. a** DEPTH scores have significant positive correlations with stemness scores, the expression levels of proliferation marker genes (*MKI67*, *TOP2A*, and *RACGAP1*), and proliferation signature scores in a wealth of validation datasets. **b** DEPTH scores have significant inverse correlations with at least three of the five immune signatures in 22 validation datasets and have significant inverse correlations with the ratios of CD8+/CD4+ regulatory T cells in 17 validation datasets (Spearman's correlation test, $p < 0.05$). **c** Kaplan−Meier survival curves displaying the negative correlation between DEPTH scores and overall survival in multiple large-scale validation datasets. The log-rank test $p$-values, HR, and 95% CI are shown. **d** The high-grade (G3-4) tumors have significantly higher DEPTH scores than the low-grade (G1-2) tumors in multiple validation datasets (one-sided Mann–Whitney U test, $p < 0.001$). **e** The AML patients with cytogenetic abnormalities (t (9;11)(p22;q23), t(10;11)(p11.2;q23), del5q, trisomy 21, or *MLL1* translocation) have significantly higher DEPTH scores than those without such abnormalities. **f** DEPTH scores show a positive correlation with the bone marrow leukemic blast percentages of AML patients. DEPTH scores are higher in the AML patients at the undifferentiated stage (M0) versus at the differentiated stage (M1-7) and are higher in the AML patients with other sites of relapse versus without other sites of relapse. **g** The high-risk cancer patients have higher DEPTH scores than low-risk cancer patients in AML and NBL. NBL, neuroblastoma. **h** Kaplan−Meier survival curves displaying the negative correlation between DEPTH scores and overall survival in pan-cancer. The results in (**e**–**h**) were obtained by analyzing the TARGET dataset.

---

by the tITH method[24], we recalculated them in the 25 TCGA cancer types based on the PPI network constructed with STRING v11[64]. The PPI network is composed of 3,284,073 edges and 15,560 genes.

**Measures of genomic instability.** We defined the TMB in a tumor sample as the total count of somatic mutations in the tumor sample. We obtained the gene mutation profiles in nine DDR pathways and HRD scores in TCGA pan-cancer from the publication[35]. We obtained the MSI data for five cancer types (COAD, ESCA, READ, STAD, and UCEC) with a high prevalence of MSI from the TCGA clinical data.

**Logistic regression analysis.** We used logistic regression with two predictors (DEPTH score and tumor purity) to predict immune signature scores and TMB (high (upper third) versus low (bottom third)), respectively. We performed the logistic regression analyses using the R function "glm" to fit the binary model and calculated the standardized regression coefficients (β values) using the function "lm.beta" in R package "QuantPsyc."

**Survival analysis.** We compared survival prognosis between the tumors with high (upper third) and low (bottom third) ITH scores. Kaplan−Meier survival curves were utilized to exhibit survival time differences. We used the log-rank test to evaluate the significance of survival time differences. We performed the survival analyses using the function "survfit" in the R package "survival." Also, we used the Cox proportional hazards model with two variables (DEPTH score and tumor purity, or DEPTH score and TILs score) to investigate the association between DEPTH score and the survival time after adjusting for the effect of tumor purity or immune signatures. Like the univariate survival analysis, we set "DEPTH score" as a binary variable with "2" (high DEPTH score (upper third)) and "1" (low (bottom third)) values. Both "tumor purity" and "TILs score" were set as continuous variables. We performed the multivariate survival analyses using the function "coxph" in R package "survival."

**Gene-set enrichment analysis.** We used GSEA[34] to identify the KEGG[65] pathways highly enriched in high-DEPTH-score (DEPTH scores in the upper third) and low-DEPTH-score (DEPTH scores in the bottom third) tumors in each cancer type using the threshold of the adjusted $p$-value (FDR) < 0.05. We defined the upper and bottom thirds in each cancer type individually, considering that different cancer types had a different distribution of gene expression values.

**Correlations of DEPTH scores with drug sensitivity.** We assessed the correlations of DEPTH scores with drug sensitivity (IC50 values) to each of the 265 compounds in 991 cancer cell lines using the Spearman's correlation test. The DEPTH scores of cancer cell lines were calculated based on their gene expression profiles without normal control samples. The significant correlations were identified using a threshold of FDR < 0.05.

**In silico simulation of tumor samples.** We performed the in silico simulation using three datasets: the 1019 human cancer cell lines from the GDSC project, and two lung cancer scRNA-seq datasets (GSE69405[28] and GSE113660[29,30]). Based on each of the three datasets, we generated a set of in silico tumor samples by randomly selecting $5 \times n$ ($n = 1, 2, 3, …, k$) cells from all cell lines or single cells to generate $k$ simulated tumor samples, where $k$ is the total number of cell lines or single cells divided by 5 (rounded). Also, based on the GDSC cell line dataset, we generated $m$ in silico tumor samples with each tumor sample composed of $m$ cell lines. The $i$th in silico tumor sample was composed of $m$ cell lines originated from $i$ different cancer types ($i = 1, 2, 3, …, m$). Using the same method, based on a glioblastoma scRNA-seq dataset (GSE57872[31]), we generated $m$ in silico tumor samples with each tumor sample composed of $m$ single cells. The $i$th in silico tumor sample was composed of $m$ single cells originated from $i$ different cell lines ($i = 1, 2, 3, …, m$).

**Statistics and reproducibility.** Unless otherwise specified, we performed two-class comparisons using the one-sided Mann–Whitney U test. The false discovery rate (FDR) was estimated by the Benjamini and Hochberg method[66] to adjust for $p$-values in multiple tests. We evaluated the correlations of ITH scores with TMB, immune signatures, proliferation signature, tumor purity, tumor stemness, tumor proliferation marker gene (*MKI67*, *TOP2A*, and *RACGAP1*) expression, and *PD-L1* expression in pan-cancer and in individual cancer types using the Spearman's correlation test. The correlation test $p$-value and correlation coefficient ($\rho$) were reported. We compared protein expression levels between high-DEPTH-score (DEPTH scores in the upper third) and low-DEPTH-score (DEPTH scores in the bottom third) tumors using Student's $t$ test. The statistically significant correlations were identified using a threshold of FDR < 0.05. The enrichment level of an immune signature in a tumor sample was the mean expression level of the marker genes of the immune signature in the tumor sample. The tumor proliferation signature and stemness scores were calculated by the single-sample gene-set enrichment analysis (ssGSEA)[67] of their marker gene sets. We defined the ratios of immune-stimulatory signature (CD8+ T cells) to immune-inhibitory signature (CD4+ regulatory T cells) as the mean expression levels of CD8+ T cell marker genes divided by the mean expression levels of CD4+ regulatory T cell marker genes (log2-transformed). We performed all the computational and statistical analyses using R (version 4.0.2) and Python (version 3.8).

**Reporting summary.** Further information on research design is available in the Nature Research Reporting Summary linked to this article.

## Data availability
The authors declare that all data supporting the findings of this study are available within the paper and its supplementary information files. The DEPTH scores of 25 TCGA patients are available at Zenodo (https://doi.org/10.5281/zenodo.3968534)[63].

## Code availability
The DEPTH algorithm (R packages) and other computer code and scripts are available at Github (https://github.com/WangX-Lab/DEPTH.git).

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

## Acknowledgements

This work was supported by the China Pharmaceutical University (grant numbers 3150120001 to XW). We thank Mr. Md. Nazim Uddin and Ms. Wenxiu Cao for their help in data collection.

## Author contributions

M.L. performed the research and data analyses and provided help in manuscript preparation and editing. Z.Z. performed the research and data analyses and provided help in manuscript preparation and editing. L.L. performed data analyses and provided help in manuscript preparation. X.W. conceived this study, designed the algorithm and analysis strategies, performed the research and data analyses, and wrote the manuscript. All the authors read and approved the final manuscript.

## Competing interests

The authors declare no competing interests.
