## [Peer Review File · Communications Biology]

Reviewers' comments:

Reviewer #1 (Remarks to the Author):

Li and colleagues have developed a method for obtaining tumor heterogeneity (TH) scores on RNA-seq sample, based on stability of tumor gene expression changes comparing with normal. They investigated if these scores are related with tumorigenesis and could be use as a proxy for clinical segregation and they compare it with other 2 DNA-based methods (ABSOLUTE and MATH).

Researchers and clinicians are accumulating lots of bulk cohorts and the development of accurate methods for deconvoluting tumor heterogeneity is necessary. The method seems simple but effective, based on the experiments performed. Nevertheless, I have some concerns regarding the comparisons:

Major concerns:

1. The authors claimed that this is the first method that quantifies TH based on mRNA, but this is not true [1,2,3]. And there are others based on epigenetics [4,5]. The whole paper is based on the comparison with 2 DNA-seq methods. It's interesting to compare how different type-based methods perform. But the main point of the paper is to highlight the greater correlations that their method yielded when comparing with the other 2. This comparison seems unfair. The authors should compare with other similar methods to demonstrate that their method is contributing with something new. Specially interesting would be comparing with [2], since both methods are based on comparable approaches.

2. Prior to the benchmark, I see necessary proof of concept of the method with some in-silico single cell simulated data, to investigate if the TH values calculated are actually related with tumor composition

3. Section 2.1: The authors compare the correlations of TMB with their method and with ABSOLUTE and MATH. They also check the correlations between TP53 mutations or MSI with DEPTH, but they do not talk about the comparison with ABSOLUTE and MATH. To keep the consistency in the text I would also talk about the comparison with the other methods.

4. Section 2.2: The pan-cancer comparison seems pointless to me. If I'm not wrong from the 25 TCGA cancer types, 5 show individual prognostic correlations with DEPTH score. I assume that these 5 types are the ones carrying the main power of the significance at pan-level. Given the heterogeneity of cancer types and the varying number of samples on each, I would skip the pan-cancer comparison and expand further on the specific ones. What about cancer subtypes (e.g: breast cancer subtypes)? Did the authors observe DEPTH scores correlating with survival in LuminalAB/Triple-/HER2 independently?

5. Section 2.2: Regarding the tumor stage comparison, same concerns as in previous point. TNM staging varies profoundly across cancer types and same stage could have different clinical implications on each cancer type. What about substages?

6. Section 2.2: About Ki67, did the authors check other proliferation markers? TOP2A? RacGAP1? Could they try with a proliferation signature?

7. Section 2.5: The combination of signatures could yield better accuracy? If the authors could include methods on other levels (epigenetics/splicing?) they could discuss on the possible advantages of using predictors relying on different data

8. Section 2.6: Assuming there is no genomic information for these datasets, it would be great if the authors could compare with other RNA based methods when performing on these microarrays

9. Section 3: Could the authors check in other data if these drug correlations persist? What if they test it in drug data on actual patients (TARGET)? This section seems interesting, but should be part of the results, not the discussion

10. Section 3: Could the authors expand on the discussion about why some cancers seem to be more correlated with RNA TH values than with DNA, and vice versa? Which cancers are more suitable for using a specific method?

11. Section 3: The discussion would be greatly improved if the comparison with other more updated and diverse methods commented in point 1 is performed. There are several methods proposing bulk tumor deconvolution. I see crucial a thorough comparison with others.

12. Section 3: the next paragraph should be stated at the beginning of results. To me it's not clear until the reader reach this part what TH is defined according to the method:

"The DEPTH algorithm shows that in a tumor, when most genes exhibit high or low expression deviations from their mean expression values in normal or tumor samples, the tumor will have a low DEPTH score (the tumor has a low heterogeneity). By contrast, when in a tumor many genes..."

Minor concern:

1. Could the method tell us what are the specific features that contributes more with TH on each cancer? In Fig4 the authors show the enriched pathways between high and low-DEPTH tumors but it would be interesting to observe some recurrence at gene level

2. SuppFig1: it would be of interest to indicate the hazard ratios and the Confidence intervals for all Kaplan-Meier curves.

3. SuppFig2: join figs A and B on a single one.

References:

1. Yoshihara K, Shahmoradgoli M, Martínez E, Vegesna R, Kim H, Torres-Garcia W, Treviño V, Shen H, Laird PW, Levine DA, Carter SL, Getz G, Stemke-Hale K, Mills GB, Verhaak R. (2013). Inferring tumour purity and stromal and immune cell admixture from expression data. *Nature Communications* 4, 2612
2. Park Y, Lim S, Nam JW, Kim S. Measuring intratumor heterogeneity by network entropy using RNA-seq data. *Sci Rep.* 2016;6:37767. Published 2016 Nov 24. doi:10.1038/srep37767
3. Kim M, Lee S, Lim S, Kim S. SpliceHetero: An information theoretic approach for measuring spliceomic intratumor heterogeneity from bulk tumor RNA-seq. *PLoS One.* 2019;14(10):e0223520. Published 2019 Oct 23. doi:10.1371/journal.pone.0223520
4. Benelli M, Romagnoli D, Demichelis F. Tumor purity quantification by clonal DNA methylation signatures. *Bioinformatics.* 2018;34(10):1642–1649. doi:10.1093/bioinformatics/bty011
5. Mazor T, Pankov A, Song JS, Costello JF. Intratumoral Heterogeneity of the Epigenome. *Cancer Cell.* 2016;29(4):440–451. doi:10.1016/j.ccell.2016.03.009

Reviewer #2 (Remarks to the Author):

Li et al. developed an algorithm (DEPTH) that quantify tumor heterogeneity based on mRNA alterations in the tumor. Using datasets from TCGA, the authors scored different types of cancer based on their algorithm and found that high DEPTH scores values show an association with

unfavorable clinical features and immunosuppressive signatures as well. Furthermore, the analysis also indicates that a high DEPTH score is associated with a low response to immunotherapy. Finally, the authors conclude that cancer prognosis and immunotherapy response could be predicted using the DEPTH score.

Major Comment:

The manuscript presents an interesting approach to quantify tumor heterogeneity and the association of this value with tumor prognosis and antitumor immunity. However, I am concerned about the reliability and robustness of the DEPTH algorithm to predict patient outcomes. There are major points in the manuscript that need to be addressed before the paper can be considered for publication.

It is not clear how the authors decided about the cancer types that were displayed in each panel of the different figures. The manuscript states that 25 cancer types were analyzed; however, only half or less are shown in the figures and not always the same cancer type. It would be interesting to consistently see the association of the DEPTH score for each cancer type with the features that the authors analyzed. As presented the data cannot be fully interpreted.

In section 2.3 of the results, the immune signature of CD8+ T cells is not properly defined, as cytotoxic CD8 T cells are not the only immune cells that express CD8a. Additional genes should be included, for example, here is a signature from GSEA for cytotoxic T cell surface molecules: CD2, CD247, CD28, CD3D, CD3E, CD3G, CD8A, ICAM1, ITGAL, ITGB2, PTPRC, THY1 at a minimum. Several other publications have described such scores ranging from 2-3 gene to ~200 genes. It would also be good to compare and contrast the findings.

Line 287: the authors state that the results from different technologies are consistent with the finding in TCGA datasets. However, in figure 6c, BRCA is the only cancer type that was also analyzed from the TCGA data set, and in figure 6d, KIRC was the only cancer analyzed using the TCGA data set. With the data provided, I am not convinced about the reliability and robustness of the DEPTH algorithm.

The discussion section needs to be improved as many of the paragraphs only repeat the main results or even present new results not mentioned in the previous section (Results) of the manuscript. Furthermore, it will be essential to integrate the results of the paper with known literature about the type of cancers that were analyzed. For example, the results in figure 1c show that high microsatellite instability (MSS) correlates with high DEPTH score in COAD, STAD, and ESCA. Is it known that these types of cancer have a high MSS? What is the DEPTH score for other types of cancer already known to have high MSS?

Line 306-307: This should be part of the result section.

Line 338-345: This should be part of the result section.

Line 347-356: This should be part of the result section. However, the rationale and the relevance of the results are not well integrated into the manuscript.

Minor comments:

In figure 3d, the colors red and blue do not match the results explained in line 222.

Line 643: The signature for CD4+ regulatory T cells should be included in the excel file (supplementary information).

Reviewer #1 (Remarks to the Author)

Major concerns:

1. The authors claimed that this is the first method that quantifies TH based on mRNA, but this is not true. And there are others based on epigenetics. The whole paper is based on the comparison with 2 DNA-seq methods. It's interesting to compare how different type-based methods perform. But the main point of the paper is to highlight the greater correlations that their method yielded when comparing with the other 2. This comparison seems unfair. The authors should compare with other similar methods to demonstrate that their method is contributing with something new. Specially interesting would be comparing with, since both methods are based on comparable approaches.

Response: We thank the reviewer for the valuable comments. Following these comments, we compared our method with other six methods for evaluating intratumor heterogeneity (ITH), including two mRNA-based methods (tITH [1] and sITH [2]) and four DNA-based methods (ABSOLUTE [3], MATH [4], EXPANDS [5], and PhyloWGS [6]). In Results section 2.10, we showed the comparison results.

2. Prior to the benchmark, I see necessary proof of concept of the method with some in-silico single cell simulated data, to investigate if the TH values calculated are actually related with tumor composition.

Response: We thank the reviewer for the valuable suggestion. Following the suggestion, we investigated DEPTH scores using in-silico single cell simulated data and demonstrated that the ITH scores calculated by DEPTH are indeed associated with tumor composition (Results section 2.11). In addition, we calculated the DEPTH scores in normal tissue whose tumor purity is supposed to be zero and found that the tumors had much higher DEPTH scores than normal tissue (Fig. 7e).

3. Section 2.1: The authors compare the correlations of TMB with their method and with ABSOLUTE and MATH. They also check the correlations between TP53 mutations or MSI with DEPTH, but they do not talk about the comparison with ABSOLUTE and MATH. To keep the consistency in the text I would also talk about the comparison with the other methods.

Response: Following the comments, we have supplemented the related comparisons, as

shown in Results section 2.10.1.

4. Section 2.2: The pan-cancer comparison seems pointless to me. If I'm not wrong from the 25 TCGA cancer types, 5 show individual prognostic correlations with DEPTH score. I assume that these 5 types are the ones carrying the main power of the significance at pan-level. Given the heterogeneity of cancer types and the varying number of samples on each, I would skip the pan-cancer comparison and expand further on the specific ones. What about cancer subtypes (e.g: breast cancer subtypes)? Did the authors observe DEPTH scores correlating with survival in LuminalAB/Triple-/HER2 independently?

Response: We thank the reviewer for the constructive comments. We agree that the prognostic correlations of DEPTH scores in pan-cancer are largely determined by the prognostic correlations in 5 individual cancer types in light of the fact that these cancer types contain a large number of tumor samples. We keep the results considering that the pan-cancer comparison is associated with four different survival endpoints. However, we think that the analysis of the correlations between DEPTH scores and clinical features in cancer subtypes as suggested by the review is a very constructive proposal. Thus, we correlated DEPTH scores with clinical features within and across subtypes of several prevalent cancer cohorts, including BRCA, COAD, LUAD, and gastrointestinal (GI) pan-cancer, and obtained many interesting results. Consistently, DEPTH scores are significantly higher in aggressive/unfavorable subtypes than in unaggressive/favorable subtypes in these cancer cohorts, e.g., in breast cancer, TNBC > non-TNBC or Basal > HER2 > Luminal AB. We presented these new results in Results section 2.2, paragraph 2&3.

5. Section 2.2: Regarding the tumor stage comparison, same concerns as in previous point. TNM staging varies profoundly across cancer types and same stage could have different clinical implications on each cancer type. What about substages?

Response: We thank the reviewer for the comments. We compared DEPTH scores between different substages in the cancers with related data available. We found that within the same stage, DEPTH scores likely increased with the advancement of substage in pan-cancer and within individual cancer types. These new results are presented in Results section 2.2, paragraph 4 and Supplementary Fig. S1c.

6. Section 2.2: About Ki67, did the authors check other proliferation markers? TOP2A? RacGAP1? Could they try with a proliferation signature?

Response: Following the suggestion, we checked the correlation between the expression of other proliferation markers (TOP2A and RacGAP1) and DEPTH scores. We observed the significant positive correlation of TOP2A and RacGAP1 expression levels with DEPTH scores in 13 and 15 cancer types, respectively. Furthermore, we obtained a proliferation signature composed of 7 marker genes [7] and found that this signature had a significant positive correlation with DEPTH scores in 21 cancer types. We presented these results in Results section 2.2, paragraph 5.

7. Section 2.5: The combination of signatures could yield better accuracy? If the authors could include methods on other levels (epigenetics/splicing?) they could discuss on the possible advantages of using predictors relying on different data.

Response: We thank the reviewer for the suggestion. We have tried to combine our method with other methods in scoring ITH and found that the combination of the different signatures could not yield better correlations between ITH and other tumor features, such as survival prognosis. However, we acknowledge that there exists certain complementarity between different ITH evaluation methods, particularly between the DNA- and mRNA-based methods for evaluating ITH. We have discussed this issue in Discussion section, paragraph 4.

8. Section 2.6: Assuming there is no genomic information for these datasets, it would be great if the authors could compare with other RNA based methods when performing on these microarrays

Response: Following this suggestion, we compared our method with another mRNA-based method tITH in the microarray datasets and presented related results in Results section 2.10.6.

9. Section 3: Could the authors check in other data if these drug correlations persist? What if they test it in drug data on actual patients (TARGET)? This section seems interesting, but should be part of the results, not the discussion

Response: Following this suggestion, we checked the correlation between DEPTH scores and drug response in five cancer patient datasets: TCGA, TARGET, GSE1379, GSE107850, and GSE123728 in addition to cancer cell lines. In general, we found that

high DEPTH scores were likely to be associated with drug resistance in these datasets. We presented these results in Results section 2.4.

10. Section 3: Could the authors expand on the discussion about why some cancers seems to be more correlated with RNA TH values than with DNA, and vice versa? Which cancers are more suitable for using a specific method?

Response: Following this suggestion, we discussed this issue in Discussion section, paragraph 4.

11. Section 3: The discussion would be greatly improved if the comparison with other more updated and diverse methods commented in point 1 is performed. There are several methods proposing bulk tumor deconvolution. I see crucial a thorough comparison with others.

Response: We thank the reviewer for this suggestion. Following the suggestion, we discussed the comparison results, as shown in Discussion section, paragraphs 2-4.

12. Section 3: the next paragraph should be stated at the beginning of results. To me it's not clear until the reader reach this part what TH is defined according to the method: "The DEPTH algorithm shows that in a tumor, when most genes exhibit high or low expression deviations from their mean expression values in normal or tumor samples, the tumor will have a low DEPTH score (the tumor has a low heterogeneity). By contrast, when in a tumor many genes..."

Response: Following this suggestion, we have moved this paragraph to the beginning of Results section.

Minor concern:

1. Could the method tell us what are the specific features that contributes more with TH on each cancer? In Fig4 the authors show the enriched pathways between high and low-DEPTH tumors but it would be interesting to observe some recurrence at gene level.

Response: We thank the reviewer for this valuable suggestion. Following this suggestion, we have identified a number of genes whose expression alteration had a strong positive correlation with DEPTH scores in at least 5 cancer types (Spearman's correlation test, $FDR < 0.05$, $\rho > 0.5$). We presented these results in Results section 2.5, paragraph 1 (Fig. 5a and Supplementary Table S2).

2. SuppFig1: it would be of interest to indicate the hazard ratios and the Confidence intervals for all Kaplan-Meier curves.

Response: Following this suggestion, we have added the hazard ratios and the Confidence intervals for all Kaplan-Meier curves.

3. SuppFig2: join figs A and B on a single one.

Response: We have done that.

References

1. Park Y, Lim S, Nam JW, Kim S. Measuring intratumor heterogeneity by network entropy using RNA-seq data. *Sci Rep.* 2016;6:37767. Epub 2016/11/25. doi: 10.1038/srep37767. PubMed PMID: 27883053; PubMed Central PMCID: PMC5121893.
2. Kim M, Lee S, Lim S, Kim S. SpliceHetero: An information theoretic approach for measuring spliceomic intratumor heterogeneity from bulk tumor RNA-seq. *PLoS One.* 2019;14(10):e0223520. Epub 2019/10/24. doi: 10.1371/journal.pone.0223520. PubMed PMID: 31644551; PubMed Central PMCID: PMC6808416.
3. Carter SL, Cibulskis K, Helman E, McKenna A, Shen H, Zack T, et al. Absolute quantification of somatic DNA alterations in human cancer. *Nat Biotechnol.* 2012;30(5):413-21. doi: 10.1038/nbt.2203. PubMed PMID: 22544022; PubMed Central PMCID: PMC4383288.
4. Mroz EA, Rocco JW. MATH, a novel measure of intratumor genetic heterogeneity, is high in poor-outcome classes of head and neck squamous cell carcinoma. *Oral Oncol.* 2013;49(3):211-5. doi: 10.1016/j.oraloncology.2012.09.007. PubMed PMID: 23079694; PubMed Central PMCID: PMC3570658.
5. Andor N, Harness JV, Muller S, Mewes HW, Petritsch C. EXPANDS: expanding ploidy and allele frequency on nested subpopulations. *Bioinformatics.* 2014;30(1):50-60. Epub 2013/11/02. doi: 10.1093/bioinformatics/btt622. PubMed PMID: 24177718; PubMed Central PMCID: PMC3866558.
6. Deshwar AG, Vembu S, Yung CK, Jang GH, Stein L, Morris Q. PhyloWGS: reconstructing subclonal composition and evolution from whole-genome sequencing of tumors. *Genome biology.* 2015;16:35. Epub 2015/03/19. doi: 10.1186/s13059-015-0602-8. PubMed PMID: 25786235; PubMed Central PMCID: PMC4359439.
7. MacDermed DM, Khodarev NN, Pitroda SP, Edwards DC, Pelizzari CA, Huang L, et al. MUC1-associated proliferation signature predicts outcomes in lung adenocarcinoma patients. *BMC medical genomics.* 2010;3:16. Epub 2010/05/13. doi: 10.1186/1755-8794-3-16. PubMed PMID: 20459602; PubMed Central PMCID: PMC2876055.

Reviewer #2 (Remarks to the Author)

Major Comment:

1. It is not clear how the authors decided about the cancer types that were displayed in each panel of the different figures. The manuscript states that 25 cancer types were analyzed; however, only half or less are shown in the figures and not always the same cancer type. It would be interesting to consistently see the association of the DEPTH score for each cancer type with the features that the authors analyzed. As presented the data cannot be fully interpreted.

Response: We thank the reviewer for this valuable suggestion. In the figures, we presented the cancer types in which the correlations were significant for DEPTH scores and did not present the cancer types in which the correlations were not significant. Because too many cancer types were analyzed, we only presented those with outstanding results to make readers more easily to catch the key points. Certainly, we think that showing the associations for each cancer type is a reasonable suggestion. In the revised manuscript, we tried to show the results for all cancer types analyzed if it is necessary.

2. In section 2.3 of the results, the immune signature of CD8+ T cells is not properly defined, as cytotoxic CD8 T cells are not the only immune cells that express CD8a. Additional genes should be included, for example, here is a signature from GSEA for cytotoxic T cell surface molecules: CD2, CD247, CD28, CD3D, CD3E, CD3G, CD8A, ICAM1, ITGAL, ITGB2, PTPRC, THY1 at a minimum. Several other publications have described such scores ranging from 2-3 gene to ~200 genes. It would also be good to compare and contrast the findings.

Response: We thank the reviewer for this valuable comment. Using the immune signature of CD8+ T cells suggested by the reviewers, we re-analyzed the correlation between CD8+ T cell enrichment scores and DEPTH scores and found that the correlation was significant in 18 cancer types. This result is similar to the previous result obtained using a single gene signature (*CD8A*).

3. Line 287: the authors state that the results from different technologies are consistent with the finding in TCGA datasets. However, in figure 6c, BRCA is the only cancer type that was also analyzed from the TCGA data set, and in figure 6d, KIRC was the only cancer analyzed using the TCGA data set. With the data provided, I am not convinced about the reliability and robustness of the DEPTH algorithm.

Response: We validated the significant correlations between DEPTH scores and immune signatures using 42 microarray datasets and the TARGET dataset. In the 42 datasets, DEPTH scores consistently showed significant positive correlations with stemness scores, the expression levels of *MKI67*, *TOP2A*, and *RACGAP1*, and proliferation signature scores in 37, 33, 37, 27, 34 datasets (Figure 9a). In 22 datasets, DEPTH scores displayed significant inverse correlations with at least 3 of the 5 immune signatures, and in 17 datasets, DEPTH scores had significant inverse correlations with CD8+/CD4+ regulatory T cells (Fig. 9b). In 11 datasets involving the tumor grade phenotype data, DEPTH scores were remarkably higher in high-grade than in low-grade tumors in 7 datasets (Figure 9d). In survival analysis, because only a few of the 42 datasets contained survival data, we reported the results in these datasets. Nevertheless, consistent with the results from the TCGA datasets, DEPTH scores were negatively associated with overall survival in several large-scale datasets (Figure 9c), in which BRCA also had a significant result in TCGA (Supplementary Figure S1a). Furthermore, we validated the findings from TCGA datasets using a new dataset TARGET (Fig. 9e-h). Overall, the results from the validation datasets are consistent with those from the TCGA datasets.

4. The discussion section needs to be improved as many of the paragraphs only repeat the main results or even present new results not mentioned in the previous section (Results) of the manuscript.

Line 306-307: This should be part of the result section.

Line 338-345: This should be part of the result section.

Line 347-356: This should be part of the result section. However, the rational and the

relevance of the results are not well integrated into the manuscript.

Response: We thank the reviewer for these valuable comments. Following the suggestions, we moved the new results presented in Discussion section to Results section: Line 306-307 to Results section 2.7; Line 338-345 to Results section 2.6; Line 347-356 to Results section 2.4.

5. Furthermore, it will be essential to integrate the results of the paper with known literature about the type of cancers that were analyzed. For example, the results in figure 1c show that high microsatellite instability (MSI) correlates with high DEPTH score in COAD, STAD, and ESCA. Is it known that these types of cancer have a high MSI? What is the DEPTH score for other types of cancer already known to have high MSI?

Response: Following the suggestion, we have integrated our results with known literature about the type of cancers that were analyzed. For example, in Results section 2.6, we linked the DEPTH score-based tumor clone numbers to those reported in the other work [8] [9]. For MSI, we analyzed 5 cancer types (COAD, STAD, ESCA, READ, and UCEC) which are known to have a high proportion of MSI tumors, and found that MSI had a significant correlation with high DEPTH score in COAD, STAD, and ESCA.

Minor comments:

1. In figure 3d, the colors red and blue do not match the results explained in line 222.

Response: We have corrected it.

2. Line 643: The signature for CD4+ regulatory T cells should be included in the excel file (supplementary information).

Response: We have corrected it.

References

8. Andor N, Graham TA, Jansen M, Xia LC, Aktipis CA, Petritsch C, et al. Pan-cancer analysis of the extent and consequences of intratumor heterogeneity. *Nature medicine*. 2016;22(1):105-13. Epub 2015/12/01. doi: 10.1038/nm.3984. PubMed PMID: 26618723; PubMed Central PMCID: PMC4830693.
9. Raynaud F, Mina M, Tavernari D, Ciriello G. Pan-cancer inference of intra-tumor heterogeneity reveals associations with different forms of genomic instability. *PLoS Genet*.

2018;14(9):e1007669. Epub 2018/09/14. doi: 10.1371/journal.pgen.1007669. PubMed PMID: 30212491; PubMed Central PMCID: PMC6155543.

Reviewers' comments:

Reviewer #1 (Remarks to the Author):

The authors have expanded the comparison with other methods. This greatly improves the benchmarking of their method and the manuscript. In addition, they have explored other possible correlations of their method with cancer subtypes and other features. Nevertheless I still see some concerns regarding the manuscript that should be reviewed prior to be considered for publication:

The results section is too long and tedious to read. Instead of giving so much details about the specific significant cancers and all the different values, I would sum up the text in just the most important results and set aside all the detailed comparisons as supplementary text. Same with figures. I would leave some as main and the rest as supplementary.

Section 2.4. The significance found when comparing the correlation of DEPTH with drug response seems misleading. This seems significant because of the great n used. No clinician would buy this.

Section 2.4. How many groups of individual drugs have been tested? Only 2 of them shown some significance? The authors state "In TARGET, we found that the acute myeloid leukemia (AML) patients with the minimal residual disease (MRD) at the end of second course of induction therapy had significantly higher DEPTH scores than those without MRD ($p = 0.059$) (Fig. 4c). Moreover, in the acute lymphoblastic leukemia (ALL) phase II patients, DEPTH scores displayed a significant positive correlation with the MRD percentages after 8 days of remission induction therapy ($p = 0.012$, $\rho = 0.25$) (Fig. 4c)." 0.059 do not seem to be very significant and $\rho = 0.25$ it's not a good correlation, specially when taking a look on the plots. The authors are overvaluing their findings and should be more cautious about their claims.

Section 2.8. It's interesting the correlations between the methods, but I would find more interesting how the other methods are correlated with all the features described in the previous methods. I think this is necessary to understand if DEPTH is actually showing new things in comparison with previous methods.

Section 2.9. Why is this comparison in a separate point? The authors are testing the correlations with clinical features and immune signatures on other datasets. This should be placed in points 2.2 and 2.3.

Section 2.10. Same concern as previous point. Why not including these comparisons on each point, instead of having an extra 2.10 point?

Section 2.10.6. Why this comparison on an extra point? Why only comparing with tITH in the validation datasets? What about the rest?

Section 2.11. This should be the first point of the results, not the last. In the methods they claim: "Based on each of the three datasets, we generated a set of in silico tumor samples by randomly selecting $5 \times n$ ($n = 1, 2, 3, \dots, k$) cells from all cell lines or single cells to generate k simulated tumor samples, where k is the total number of cell lines or single cells divided by 5 (rounded)." According to this, I assume that the only tumor with 100% cells from only one cell line is $k=1$. Then the following tumors have 5 cells from $n=1,2,3\dots$ cell lines, am I wrong? This experiment is interesting and shows the correlation between tumor complexity and DEPTH score. I think it would give more credit adding an experiment in which the different simulated samples are controlled by the amount of heterogeneity on each (e.g: from fixed number of n cells, go from pure samples (all cells from same cell line) to superheterogenous samples (cells from many cell lines)) and checking if the DEPTH score actually correlates with the entropy (Shannon index?) of the population. Create m samples for each step of heterogeneity for assuring the stability of the method

Section 3. Paragraph 4: The authors comment the differences between the correlations of the different ITH values. They observed that for some cancer DEPTH score has greater correlations than the rest of the methods and viceversa. In my previous concerns, I was suggesting if a combined approach could show better results than only relying on independent ITH values.

Section 3. The authors do not mention anything about sITH method, that shows good correlations regarding the genomic instability (TP53, HRD) or clinical features (stage).

Reviewer #2 (Remarks to the Author):

While the authors have addressed the specific concerns raised it is still unclear to the reviewer how specifically their method is different to other prediction-methods such as simple gene expression correlation. From my perspective the advances for the field would be minimal.

Responses to reviewers' comments

Reviewer #1 (Remarks to the Author):

1) The results section is too long and tedious to read. Instead of giving so much details about the specific significant cancers and all the different values, I would sum up the text in just the most important results and set aside all the detailed comparisons as supplementary text. Same with figures. I would leave some as main and the rest as supplementary.

Response: We thank the reviewer for the constructive comments. Following the comments, we have moved some data to the Supplementary Information, such as the data on the comparisons of DEPTH with other ITH evaluation methods.

2) Section 2.4. The significance found when comparing the correlation of DEPTH with drug response seems misleading. This seems significant because of the great n used. No clinician would buy this.

Response: We thank the reviewer for the valuable comments. We first analyzed the correlation between DEPTH scores and drug response in cancer cell lines based on the Genomics of Drug Sensitivity in Cancer (GDSC) dataset. We found that DEPTH scores had significant correlations with drug sensitivity (IC50 values) to 180 (68%) of 265 compounds tested in cancer cell lines. We analyzed the correlation between DEPTH scores and drug response in cancer patients using TCGA, TARGET, and three GEO (GSE1379, GSE107850, and GSE123728) datasets. In these datasets, DEPTH scores overall displayed a significant association with drug response in cancer. For example, in all three GEO datasets, the DEPTH score was associated with the sensitivity to anticancer drugs in cancer patients. Nevertheless, we agree that the correlation between DEPTH scores and drug response needs more validation considering that in some datasets, such as TCGA, a number of drugs were tested.

3) Section 2.4. How many groups of individual drugs have been tested? Only 2 of them shown some significance? The authors state "In TARGET, we found that the acute myeloid leukemia (AML) patients with the minimal residual disease (MRD) at the end of second course of induction therapy had significantly higher DEPTH scores than those without MRD ($p = 0.059$) (Fig. 4c). Moreover, in the acute lymphoblastic leukemia (ALL) phase II patients, DEPTH scores

displayed a significant positive correlation with the MRD percentages after 8 days of remission induction therapy ($p = 0.012$, $\rho = 0.25$) (Fig. 4c)." 0.059 do not seem to be very significant and $\rho = 0.25$ it's not a good correlation, specially when taking a look on the plots. The authors are overvaluing their findings and should be more cautious about their claims.

Response: We thank the reviewer for the valuable comments. In TARGET, a total of 9 groups of individual drugs have been tested, of which 2 groups showed specific significance. We agree that these findings should be interpreted cautiously.

4) Section 2.8. It's interesting the correlations between the methods, but I would find more interesting how the other methods are correlated with all the features described in the previous methods. I think this is necessary to understand if DEPTH is actually showing new things in comparison with previous methods.

Response: We thank the reviewer for the valuable comments. We have analyzed the correlations between the other methods and the common features (such as genomic instability, unfavorable clinical features, antitumor immunosuppression, drug resistance, and tumor purity) of intratumor heterogeneity (ITH) and compared the correlations between our method and the other methods (see Supplementary Information). Our data shows that DEPTH scores tended to display a stronger and more consistent correlation with these features than the ITH scores derived from the other methods in TCGA pan-cancer and 25 individual cancer types. Compared to the two mRNA-based methods, DEPTH scores show stronger and more consistent associations with genomic instability, unfavorable clinical features, and drug resistance. DEPTH scores can capture genetic alterations in cancer to a higher degree than both methods in that DEPTH scores show a stronger correlation with the genomic instability features, including tumor mutation burden (TMB), *TP53* mutations, and homologous recombination deficiency. Compared to the DNA-based ITH scores, DEPTH scores show a stronger correlation with immune signatures, cell proliferation, stemness, tumor progression, prognosis, and even TMB. Notably, unlike the other methods, DEPTH can evaluate ITH in the case of gene expression profiles in normal samples being unavailable.

5) Section 2.9. Why is this comparison in a separate point? The authors are testing the correlations with clinical features and immune signatures on other datasets. This should be placed in points 2.2 and 2.3.

Response: We thank the reviewer for the valuable suggestion. We put this section separately because we considered it as an independent validation of our previous findings in the TCGA datasets. The validation set included 42 microarray gene expression profiling datasets generated by different technologies and platforms and containing different numbers of genes. To highlight the DEPTH algorithm's reliability and robustness, we validated the correlations between DEPTH scores and clinical features and immune signatures on these datasets in a separate point.

6) Section 2.10. Same concern as previous point. Why not including these comparisons on each point, instead of having an extra 2.10 point?

Response: As suggested by the reviewers, a sufficient comparison of DEPTH with other ITH evaluation algorithms is necessary to improve the manuscript. Following the suggestion, we have performed a thorough comparison of DEPTH with the other methods. Because there were six methods and five points for comparison, we specially included these comparisons on each point in an individual subsection to demonstrate them. However, as suggested by the reviewer, we moved these data to the Supplementary Information to avoid the main text's tediousness.

7) Section 2.10.6. Why this comparison on an extra point? Why only comparing with tITH in the validation datasets? What about the rest?

Response: Among the six methods compared with DEPTH, tITH is the only one that was developed based on the gene expression profiles in cancer, as is the same as DEPTH. Because all validation datasets were microarray gene expression profiling datasets, we could not evaluate the ITH in these datasets using the other algorithms except tITH and DEPTH. Furthermore, as pointed by the reviewer previously, an extensive comparison of DEPTH with other mRNA-based methods is essential. Thus, we put the comparison of DEPTH with tITH in the validation datasets in an individual subsection.

8) Section 2.11. This should be the first point of the results, not the last. In the methods they claim: "Based on each of the three datasets, we generated a set of in silico tumor samples by randomly selecting $5 \times n$ ($n = 1, 2, 3, \dots, k$) cells from all cell lines or single cells to generate k simulated tumor samples, where k is the total number of cell lines or single cells divided by 5 (rounded)."

According to this, I assume that the only tumor with 100% cells from only one cell line is $k=1$. Then the following tumors have 5 cells from $n=1,2,3\dots$ cell lines, am I wrong? This experiment is interesting and shows the correlation between tumor complexity and DEPTH score. I think it would give more credit adding an experiment in which the different simulated samples are controlled by the amount of heterogeneity on each (e.g: from fixed number of n cells, go from pure samples (all cells from same cell line) to super heterogenous samples (cells from many cell lines)) and checking if the DEPTH score actually correlates with the entropy (Shannon index?) of the population. Create m samples for each step of heterogeneity for assuring the stability of the method

Response: This a valuable suggestion. Following the suggestion, we have supplemented the related experiment in Section 2.1 (we also moved this section to the first point of Results as suggested by the reviewer). The new experiment showed that the more heterogeneous tumor samples (composed of more different types of cell lines or cancers) had higher DEPTH scores, demonstrating that DEPTH scores indeed represent ITH in tumors.

9) Section 3. Paragraph 4: The authors comment the differences between the correlations of the different ITH values. They observed that for some cancer DEPTH score has greater correlations than the rest of the methods and vice versa. In my previous concerns, I was suggesting if a combined approach could show better results than only relying on independent ITH values.

Response: We thank the reviewer for the valuable suggestion. Following the suggestion, we have tested the combination of DEPTH and the other ITH methods to evaluate the ITH. However, we did not observe significantly better results. For example, we have used the cox proportional hazards model to correlate the combination of the DEPTH ITH values and the other method's ITH values with survival, but we did not obtain better results than those based on independent ITH values. However, we agree that the combined approach has the potential to improve the evaluation of ITH.

10) Section 3. The authors do not mention anything about sITH method that shows good correlations regarding the genomic instability (TP53, HRD) or clinical features (stage).

Response: We thank the reviewer for this comment. Although sITH is an mRNA-based ITH evaluation method, it measures the spliceomic ITH, but does not measure the gene expression profiling ITH. Thus, we did not focus on it but more focused on tITH. Indeed, the sITH showed

good correlations with genomic instability (such as *TP53* mutations and HRD) or certain clinical features (such as stage). We have also compared these correlations between DEPTH and sITH. Another interesting finding was that the correlation between DEPTH and sITH was stronger than that between tITH and sITH, although both tITH and sITH were developed based on the Jensen-Shannon Divergence measure.

Reviewer #2 (Remarks to the Author):

While the authors have addressed the specific concerns raised it is still unclear to the reviewer how specifically their method is different to other prediction-methods such as simple gene expression correlation. From my perspective the advances for the field would be minimal.

Response: We thank the reviewer for the comments. The intratumor heterogeneity (ITH) level is a biomarker of cancer prognosis and therapy response. In this study, we proposed the DEPTH algorithm, a new mRNA-based method for evaluating ITH. DEPTH evaluates the ITH based on the asynchrony of transcriptome alterations in tumor cells. The novel idea behind this algorithm is concise but effective. We believe that the DEPTH algorithm is valuable. First, most of the previous studies evaluating ITH levels were based on DNA alterations, and very few were based on mRNA alterations. Second, the DEPTH algorithm exhibits certain superiorities in quantifying ITH. By comparing DEPTH with other ITH evaluation methods, we proved that DEPTH is superior to or comparable the other methods in that the DEPTH ITH tends to display a stronger and more consistent correlation with the common features of the ITH (such as genomic instability, tumor advancement, unfavorable prognosis, immunosuppression, and drug resistance) than the ITH derived from the other methods. One prominent advantage of DEPTH over the other ITH evaluation algorithms is that DEPTH can evaluate ITH in the case of gene expression profiles in normal samples being unavailable. Thus, we believe that DEPTH is valuable for advancing our understanding of tumor heterogeneity, an important topic in tumor biology.

REVIEWERS' COMMENTS:

Reviewer #1 (Remarks to the Author):

The authors have improved the manuscript. They have made a great effort to give an extensive analysis on the relevance of the value their method calculates and how related is this with several clinical measures. I still has some concerns about the text:

Major

I already mention this but the authors are basing many of the findings in the manuscript on the significance of their correlations. While in many cases these associations seems interesting in some others the correlations are really poor, and given the high number of samples they are dealing with it's likely to found significance. This doesn't mean that there is an actual correlation.

Section 2.2: Are the DEPTH scores comparable between experiments?

Section 2.3: About the LUAD analysis: the EGFR mutation association with favorable prognosis is in patients with brain metastasis. I assume these analysys were performed on TCGA. Are these samples with a brain metastasis?

Fig3a: How are the low and high score groups splitted? The authors should give a little bit more explanation on the analysis performed.

Fig4: On the legend is mentioned: [...] (b) The significant inverse correlations of DEPTH scores with the ratios of immune-stimulatory signature (CD8+ T cells) to immune-inhibitory signature (CD4+ regulatory T cells) in pan-cancer and in 12 individual cancer types (FDRsp < 0.05). [...] In the fig there are more than 12 cancer types, which are the ones that the authors considered as significantly associated?

Fig 4d, 5d: would be better showing the dot distribution instead of bar plots

Minor:

Suppl fig 2: Review the legend. In the text a 0.02 value is mentioned, whereas in the legend it shows 0.05.

Responses to reviewers' comments

Reviewer #1 (Remarks to the Author):

1) I already mention this but the authors are basing many of the findings in the manuscript on the significance of their correlations. While in many cases these associations seem interesting in some others the correlations are really poor, and given the high number of samples they are dealing with it's likely to found significance. This doesn't mean that there is an actual correlation.

Response: We thank the reviewer for the comments. We acknowledge that some correlations between DEPTH scores and clinical parameters were affected by the large number of samples and thus they need to cautiously considered. Nevertheless, we believe that most of these associations related to tumor heterogeneity are meaningful.

2) Section 2.2: Are the DEPTH scores comparable between experiments?

Response: This is an excellent question. Making the DEPTH scores comparable between different experimental data is crucial for its applicability in the research and even clinical settings. To do this, the normalization of gene experimental data before calculating DEPTH scores is important. Some commonly-used normalization methods can be used, such as quantile or global normalization, z-score, and min-max normalization. We will intensively investigate this issue in the future study.

3) Section 2.3: About the LUAD analysis: the EGFR mutation association with favorable prognosis is in patients with brain metastasis. I assume these analyses were performed on TCGA. Are these samples with a brain metastasis?

Response: Yes, we performed the LUAD analysis on TCGA. In the TCGA LUAD cohort, 77 of the 490 tumor samples were with brain metastasis, including 11 EGFR-mutated and 66 EGFR-wildtype samples. Actually, the association between EGFR mutations and favorable prognosis in LUAD patients is not restricted in LUAD with brain metastasis. We have added a new reference (Ref. [36]) related to this fact.

4) Fig3a: How are the low and high score groups splitted? The authors should give a little bit more explanation on the analysis performed.

Response: We thank the reviewer for the comment. The high-DEPTH-score group included the tumors with DEPTH scores in the upper third, and the low-DEPTH-score group included the tumors with DEPTH scores in the bottom third. The upper and bottom thirds were calculated in each cancer type or pan-cancer individually considering that different cancer types had different distribution of gene expression values. We have added an explanation on this in the Methods section.

5) Fig4: On the legend is mentioned: [...] (b) The significant inverse correlations of DEPTH scores with the ratios of immune-stimulatory signature (CD8+ T cells) to immune-inhibitory signature (CD4+ regulatory T cells) in pan-cancer and in 12 individual cancer types (FDRsp < 0.05). [...]

In the fig there are more than 12 cancer types, which are the ones that the authors considered as significantly associated?

Response: We thank the reviewer for the comment. We have marked the cancer types with significant correlations using a special character, as shown in Fig. 4b.

6) Fig 4d, 5d: would be better showing the dot distribution instead of bar plots

Response: Following this suggestion, we did dot distribution plots instead of bar plots in Figs 4d, 5d.

7) Suppl fig 2: Review the legend. In the text a 0.02 value is mentioned, whereas in the legend it shows 0.05.

Response: We have changed “0.05” into “0.02” in the legend.